# Understanding Fairness and Prediction Error through Subspace Decomposition and Influence Analysis

**Enze Shi, Pankaj Bhagwat, Zhixian Yang, Linglong Kong, Bei Jiang**[*]

Department of Mathematical and Statistical Science
University of Alberta
{eshi,pbhagwat,zhixian,lkong,bei1}@ualberta.ca

## Abstract

Machine learning models have achieved widespread success but often inherit and amplify historical biases, resulting in unfair outcomes. Traditional fairness methods typically impose constraints at the prediction level, without addressing underlying biases in data representations. In this work, we propose a principled framework that adjusts data representations to balance predictive utility and fairness. Using sufficient dimension reduction, we decompose the feature space into target-relevant, sensitive, and shared components, and control the fairness–utility trade-off by selectively removing sensitive information. We provide a theoretical analysis of how prediction error and fairness gaps evolve as shared subspaces are added, and employ influence functions to quantify their effects on the asymptotic behavior of parameter estimates. Experiments on both synthetic and real-world datasets validate our theoretical insights and show that the proposed method effectively improves fairness while preserving predictive performance.

## 1 Introduction

Machine learning (ML) models have achieved remarkable success across a wide range of high-stakes applications, including finance Hardt et al. [2016], Liu et al. [2018], healthcare Potash et al. [2015], Rudin and Ustun [2018], and criminal justice Van Dijck [2022], Billi et al. [2023]. Despite these advances, growing evidence highlights that ML systems often inherit and reinforce historical biases, leading to unfair outcomes Tolan et al. [2019], Mehrabi et al. [2021]. Biases in data collection Liang et al. [2020], Pagano et al. [2023] and disparities in group representation De-Arteaga et al. [2019], Dablain et al. [2024] can manifest in model predictions, ultimately amplifying social inequities Bolukbasi et al. [2016], Hassani [2021], Hu et al. [2024], Ding et al. [2024].

To address fairness concerns, researchers have introduced a range of formal definitions and algorithmic interventions. Early work focused on ensuring statistical criteria such as Demographic Parity Kamishima et al. [2012], Jiang et al. [2020], and more recent developments extend fairness guarantees to multiple sensitive attributes Tian et al. [2024], Chen et al. [2024] or local prediction regions Jin et al. [2024]. Many approaches operationalize fairness by adding constraints or regularization terms to learning objectives Hardt et al. [2016], Li et al. [2023]. However, this strategy typically treats fairness as an external correction layered on top of predictive modeling, without addressing the root causes of bias encoded within data representations themselves.

This work proposes a new perspective: instead of adjusting model predictions post hoc, we seek to understand and manage the trade-off between utility and fairness at the level of data representation. Specifically, we focus on how to construct representations that balance predictive accuracy for the

---

[*]Corresponding Author

39th Conference on Neural Information Processing Systems (NeurIPS 2025).

target variable and independence from sensitive attributes. This viewpoint naturally connects fairness with the framework of sufficient dimension reduction (SDR) Cook and Li [2002], Adragni and Cook [2009], where the goal is to project high-dimensional features onto lower-dimensional subspaces that preserve essential information.

We consider a structured setup in which both the prediction target $Y$ and the sensitive attribute $Z$ admit low-dimensional sufficient reductions with respect to the covariates $X$. By analyzing the subspaces associated with $Y$ and $Z$, we separate two types of directions: (1) directions informative about $Y$ but orthogonal to $Z$ and (2) directions jointly informative about both $Y$ and $Z$. This decomposition enables a principled subspace selection: start with $Z$-orthogonal directions for fairness, then add shared ones to boost accuracy as needed.

To quantify the fairness–utility trade-off, we develop a theoretical framework showing how prediction error and group-wise disparities evolve as shared directions are added. While incorporating $Z$-related directions improves accuracy, it reintroduces unfairness, which is captured through variance decomposition and gap metrics. We further apply influence functions to analyze the impact of partially fair representations on parameter estimation and prediction. The contributions of this paper are summarized as follows:

1. We propose a general framework for fairness-aware learning by directly manipulating data representations. It enables controlled removal of sensitive information without relying on a specific fairness definition.
2. We theoretically characterize how prediction risk and fairness gaps evolve as shared information is gradually introduced, and use influence functions to analyze the effect of partially fair representations on estimation and prediction.
3. We validate our approach through experiments on synthetic and real-world datasets, demonstrating that subspace elimination achieves trade-offs between fairness and predictive performance.

The structure of this paper is as follows. Section 3 introduces the motivation and problem setup, along with theoretical analysis of the utility–fairness trade-off under subspace elimination. Section 4 details the estimation procedure. Section 5 analyzes the asymptotic behavior of the estimators and predictors using influence functions. Section 6 presents simulation and real-world experiments that demonstrate the effectiveness of our approach.

## 2 Related Work

A major approach to fairness in machine learning focuses on enforcing fairness during model training, known as in-processing Wang et al. [2022a], Berk et al. [2023], Caton and Haas [2024]. These methods incorporate fairness constraints Zafar et al. [2019], Agarwal et al. [2019] into the optimization objective to satisfy predefined criteria such as Demographic Parity Dwork et al. [2012] or Equality of Opportunity Hardt et al. [2016], Shen et al. [2022], and have been applied across various paradigms including contrastive learning Wang et al. [2022b], Zhang et al. [2022], adversarial training Han et al. [2021], domain adaptation Wang et al. [2020], and balanced supervised learning Han et al. [2022]. However, fairness guarantees obtained this way are often task-specific and may not generalize across downstream applications. A complementary direction focuses on achieving fairness at the representation level by removing sensitive information from feature embeddings. Notable methods such as Iterative Null-space Projection (INLP) Ravfogel et al. [2020, 2023] and Relaxed Linear Adversarial Concept Erasure (RLACE) Ravfogel et al. [2022] aim to systematically eliminate sensitive information. More recently, sufficient dimension reduction (SDR) techniques have been adapted for debiasing Shi et al. [2024], identifying and removing subspaces associated with sensitive attributes while offering theoretical guarantees, with an extension of non-linear SDR via deep neural networks as proposed in Shi et al. [2025].

## 3 Methodology

### 3.1 Motivation

We study the problem of removing sensitive information from vector representations while preserving task-relevant content. Let $(X, Y, Z)$ be random variables where $X \in \mathbb{R}^p$ denotes the representation,

$\boldsymbol{Y} \in \mathbb{R}^K$ is the prediction target, and $\boldsymbol{Z} \in \mathbb{R}^d$ is the sensitive attribute. Our goal is to learn a transformation $h : \mathbb{R}^p \to \mathbb{R}^p$ such that the transformed representation $h(\boldsymbol{X})$ satisfies: (1) Minimal dependence on the sensitive attribute $\boldsymbol{Z}$ and (2) Sufficient information for predicting the target variable $\boldsymbol{Y}$. A direct approach involves constructing a linear transformation $h(\boldsymbol{X}) = P\boldsymbol{X}$, where $P \in \mathbb{R}^{p \times p}$ is a projection matrix. This induces a decomposition of the input space as $\boldsymbol{X} = P\boldsymbol{X} + (I - P)\boldsymbol{X}$, with $P$ projecting onto a subspace $\mathcal{S}_1 \subseteq \mathbb{R}^p$ and $I - P$ projecting onto its orthogonal complement $\mathcal{S}_2 = \mathcal{S}_1^\perp$. The representation space is thus decomposed as $\mathbb{R}^p = \mathcal{S}_1 \oplus \mathcal{S}_2$. For effective debiasing, $\mathcal{S}_1$ should minimize information about $\boldsymbol{Z}$, while $\mathcal{S}_2$ captures the $\boldsymbol{Z}$-relevant components to be removed.

This formulation is introduced by Shi et al. [2024], who adopt the sufficient dimension reduction (SDR) framework to identify and eliminate sensitive information in representations. Given $\boldsymbol{X}$ and $\boldsymbol{Z}$, their goal is to find a matrix $B_z \in \mathbb{R}^{p \times r}$ with orthonormal columns such that

$$\boldsymbol{Z} \perp\!\!\!\perp \boldsymbol{X} \mid B_z^\top \boldsymbol{X}, \tag{1}$$

ensuring that $B_z^\top \boldsymbol{X}$ contains all the information in $\boldsymbol{X}$ relevant to predicting $\boldsymbol{Z}$. The column space of $B_z$, known as the SDR subspace of $\boldsymbol{X}$ with respect to $\boldsymbol{Z}$, see Cook and Li [2002], Adragni and Cook [2009], thus serves as a natural candidate for $\mathcal{S}_2$ with $\dim(\mathcal{S}_2) = r$. Its orthogonal complement, spanned by $P = I - B_z B_z^\top$, defines $\mathcal{S}_1$ with $\dim(\mathcal{S}_1) = p - r$. Then $h(\boldsymbol{X}) = P\boldsymbol{X}$ removes information associated with $\boldsymbol{Z}$, yielding a fair projection.

While the sufficient projection method in Shi et al. [2024] performs well across downstream tasks, it primarily focuses on removing $\boldsymbol{Z}$-related information without explicitly preserving information relevant to $\boldsymbol{Y}$. This can lead to substantial utility loss when $\boldsymbol{Y}$ and $\boldsymbol{Z}$ share overlapping subspaces. To address this, we propose a finer decomposition of the representation space $\mathbb{R}^p$, separating directions informative about $\boldsymbol{Y}$ but orthogonal to $\boldsymbol{Z}$, directions shared by both, and directions unrelated to either. This more granular perspective allows for better control of the fairness–utility trade-off by retaining task-relevant features while minimizing bias from sensitive attributes.

### 3.2 Problem Setup

We adopt a SDR framework for both the target variable $\boldsymbol{Y}$ and the sensitive attribute $\boldsymbol{Z}$, modeled as

$$\boldsymbol{Y} = f(\beta_1^\top \boldsymbol{X}, \beta_2^\top \boldsymbol{X}, \dots, \beta_q^\top \boldsymbol{X}, \varepsilon_Y), \tag{2}$$

$$\boldsymbol{Z} = g(\psi_1^\top \boldsymbol{X}, \psi_2^\top \boldsymbol{X}, \dots, \psi_r^\top \boldsymbol{X}, \varepsilon_Z), \tag{3}$$

for some measurable functions $f$ and $g$, where $\{\beta_k\}_{k=1}^q \subset \mathbb{R}^p$ and $\{\psi_j\}_{j=1}^r \subset \mathbb{R}^p$ are orthonormal direction vectors, and $\varepsilon_Y$, $\varepsilon_Z$ are noise terms independent of $\boldsymbol{X}$. The SDR assumption implies that all information relevant for predicting $\boldsymbol{Y}$ and $\boldsymbol{Z}$ is captured by the low-dimensional projections $\{\beta_k^\top \boldsymbol{X}\}_{k=1}^q$ and $\{\psi_j^\top \boldsymbol{X}\}_{j=1}^r$, respectively. Equivalently, the models in (2) and (3) imply the conditional independence statements:

$$\boldsymbol{Y} \perp\!\!\!\perp \boldsymbol{X} \mid M_Y \boldsymbol{X}, \quad \boldsymbol{Z} \perp\!\!\!\perp \boldsymbol{X} \mid M_Z \boldsymbol{X}, \tag{4}$$

where $M_Y$, $M_Z \in \mathbb{R}^{p \times p}$ are matrices with rank $q$ and $r$. The subspaces spanned by these matrices are referred to as the central subspaces, defined by

$$\mathcal{S}_{\boldsymbol{Y}|\boldsymbol{X}} = \mathrm{Span}(M_Y) = \mathrm{Span}\{\beta_1, \dots, \beta_q\}, \quad \mathcal{S}_{\boldsymbol{Z}|\boldsymbol{X}} = \mathrm{Span}(M_Z) = \mathrm{Span}\{\psi_1, \dots, \psi_r\}.$$

Assume that the central subspaces $\mathcal{S}_{\boldsymbol{Y}|\boldsymbol{X}}$ and $\mathcal{S}_{\boldsymbol{Z}|\boldsymbol{X}}$ intersect in a subspace

$$\mathcal{S}_{\boldsymbol{Y}|\boldsymbol{X}} \cap \mathcal{S}_{\boldsymbol{Z}|\boldsymbol{X}} = \mathrm{Span}\{\phi_1, \dots, \phi_s\},$$

where $s \le \min\{q, r\}$. When $s = 0$, the subspaces intersect only at the origin. In the special case where $\mathcal{S}_{\boldsymbol{Y}|\boldsymbol{X}} \subset \mathcal{S}_{\boldsymbol{Z}|\boldsymbol{X}}^\perp$, the two subspaces are completely separable. In this setting, removing all information associated with $\boldsymbol{Z}$ does not affect the information relevant for predicting $\boldsymbol{Y}$, and thus fairness can be achieved without sacrificing utility.

In practice, the subspaces $\mathcal{S}_{\boldsymbol{Y}|\boldsymbol{X}}$ and $\mathcal{S}_{\boldsymbol{Z}|\boldsymbol{X}}$ often overlap, with a nontrivial intersection ($s > 0$). In such cases, removing all $\boldsymbol{Z}$-related components may also eliminate valuable information for predicting $\boldsymbol{Y}$. To address this, we decompose $\mathcal{S}_{\boldsymbol{Y}|\boldsymbol{X}}$ into two orthogonal parts: one shared with $\mathcal{S}_{\boldsymbol{Z}|\boldsymbol{X}}$ and one independent of it. This decomposition enables a principled approach to balancing fairness and utility by selectively retaining target-relevant features uncorrelated with the sensitive attribute.

Without loss of generality, we assume that the shared basis vectors satisfy $\phi_i = \beta_{q-s+i} = \psi_{r-s+i}$ for $i = 1, \ldots, s$. Define the projection matrix onto $\mathcal{S}_{\boldsymbol{Z}|\boldsymbol{X}}$ as $P_z = \sum_{j=1}^r \psi_j \psi_j^\top$ and $Q_z = I_p - P_z$, where $Q_z$ projects onto the orthogonal complement of the sensitive subspace. Let $B = (\beta_1, \ldots, \beta_q) \in \mathbb{R}^{p \times q}$ and $\Phi = (\phi_1, \ldots, \phi_s) \in \mathbb{R}^{p \times s}$. Since $\mathcal{S}_{\boldsymbol{Y}|\boldsymbol{X}} \cap \mathcal{S}_{\boldsymbol{Z}|\boldsymbol{X}} = \mathrm{Span}(\Phi)$, the uncorrelated component $\mathcal{S}_{\boldsymbol{Y}|\boldsymbol{X}} \cap \mathcal{S}_{\boldsymbol{Z}|\boldsymbol{X}}^\perp$ can be identified by finding a matrix $\widetilde{B}$ such that

$$\boldsymbol{Y} \perp\!\!\!\perp Q_z \boldsymbol{X} \mid \widetilde{B}^\top \boldsymbol{X}, \tag{5}$$

**Theorem 3.1.** *Let $\widetilde{B}$ be an orthonormal matrix satisfying condition* (5)*, then* $\mathrm{Span}(\widetilde{B}) \subseteq \mathrm{Span}(Q_z B) = \mathcal{S}_{\boldsymbol{Y}|\boldsymbol{X}} \cap \mathcal{S}_{\boldsymbol{Z}|\boldsymbol{X}}^\perp$.

Theorem 3.1 provides a direct link between the central subspace of $\boldsymbol{Y}$ and its component orthogonal to the sensitive subspace $\mathcal{S}_{\boldsymbol{Z}|\boldsymbol{X}}$. Building on this decomposition, we define a sequence of partially fair projection matrices $\{P^{(m)}\}_{m=0}^s$ as

$$P^{(m)} = \widetilde{B}\widetilde{B}^\top + \Phi_m \Phi_m^\top, \quad \text{where} \quad \Phi_m = (\phi_1, \ldots, \phi_m).$$

Each $P^{(m)}$ projects $\boldsymbol{X}$ onto a subspace that retains $\widetilde{B}$-based directions uncorrelated with $\boldsymbol{Z}$, along with $m$ of the $s$ shared directions between $\boldsymbol{Y}$ and $\boldsymbol{Z}$. Let $\boldsymbol{\Xi}^{(m)} = P^{(m)} \boldsymbol{X}$ denote the resulting representation. When $m = 0$, the representation is entirely uncorrelated with $\boldsymbol{Z}$; when $m = s$, the representation spans the full central subspace of $\boldsymbol{Y}$, preserving complete utility but potentially reintroducing bias.

To study the predictive behavior under this fairness–utility trade-off, we define the Bayes optimal predictor using the partially fair representation:

$$\tilde{f}^{(m)}(\boldsymbol{\Xi}^{(m)}) = \mathbb{E}[\boldsymbol{Y} \mid \boldsymbol{\Xi}^{(m)}].$$

This formulation allows gradual control over the balance between fairness and accuracy by varying $m$, i.e., the number of shared components included. In the following, we analyze the theoretical properties of $\tilde{f}^{(m)}$, characterizing how prediction risk and fairness evolve with subspace selection.

### 3.3 Utility and Fairness Trade-off

Let $\boldsymbol{\Xi}^{(m)} = P^{(m)} \boldsymbol{X}$ denote the reduced representation. The following result characterizes the prediction error of the Bayes optimal predictor based on the reduced representation.

**Theorem 3.2.** *Let $\tilde{f}^{(m)}(\boldsymbol{\Xi}^{(m)}) = \mathbb{E}[\boldsymbol{Y} \mid \boldsymbol{\Xi}^{(m)}]$ denote the Bayes predictor using the partially fair representation $\boldsymbol{\Xi}^{(m)}$, and let $f^*(\boldsymbol{X}) = \mathbb{E}[\boldsymbol{Y} \mid \boldsymbol{X}]$ denote the Bayes optimal predictor using the original representation. Then, the expected squared prediction error satisfies*

$$\mathbb{E}[(\boldsymbol{Y} - \tilde{f}^{(m)}(\boldsymbol{\Xi}^{(m)}))^2] = \underbrace{\mathbb{E}[\mathrm{Var}(f^*(\boldsymbol{X}) \mid \boldsymbol{\Xi}^{(m)})]}_{\textit{Approximation error}} + \underbrace{\mathbb{E}[\varepsilon_Y^2]}_{\textit{Irreducible noise}} := \Delta(m) + \sigma_Y^2.$$

*Moreover, the approximation error $\Delta(m)$ is non-increasing in $m$, i.e.,*

$$\Delta(m+1) \leq \Delta(m) \quad \textit{for all } m \in \{0, \ldots, s-1\}.$$

This decomposition follows from the orthogonality principle in $L^2$ space, which ensures that the conditional expectation minimizes mean squared error. The term $\mathbb{E}[\mathrm{Var}(f^*(\boldsymbol{X}) \mid \boldsymbol{\Xi}^{(m)})]$ quantifies the loss in predictive information due to reducing the representation to $\boldsymbol{\Xi}^{(m)}$. As more shared directions are included, the reduced representation becomes increasingly informative, and the error $\Delta(m)$ decreases. When $m = s$, the full central subspace for $\boldsymbol{Y}$ is recovered, yielding $\Delta(s) = 0$.

Quantifying unfairness theoretically is inherently challenging, as it often stems from disparities in prediction outcomes across sensitive subpopulations. Instead of relying on specific fairness metrics like TPR or demographic parity (DP) gaps, we adopt a distributional perspective by measuring the statistical dependence between the predictor and the sensitive attribute using *distance covariance* (dCov) Székely et al. [2007], which quantifies the discrepancy between joint and marginal characteristic functions. The following result illustrates how the reduced representation $\boldsymbol{\Xi}^{(m)}$ mitigates unfairness by weakening the dependency between the predictor and the sensitive attribute. While the result is stated for binary $\boldsymbol{Z}$, it naturally extends to multivariate cases with $\boldsymbol{Z} \in \mathbb{R}^d$.

**Theorem 3.3.** *Let $Z \in \{0, 1\}$ be a binary sensitive attribute with $p = \mathbb{P}(Z = 1)$, and let $\tilde{f}^{(m)} = \mathbb{E}[\boldsymbol{Y} \mid \boldsymbol{\Xi}^{(m)}]$ be the Bayes predictor using the reduced SDR representation $\boldsymbol{\Xi}^{(m)}$. Then the squared population distance covariance between $\tilde{f}^{(m)}$ and $Z$ satisfies*

$$\mathrm{dCov}^2(\tilde{f}^{(m)}, Z) = 2p(1-p)\left(\mathbb{E}[|\tilde{f}_1^{(m)} - \tilde{f}_0^{(m)}|] - \mathbb{E}[|\tilde{f}^{(m)} - \tilde{f}^{(m)'}|]\right),$$

*where $\tilde{f}_z^{(m)} \sim \tilde{f}^{(m)} \mid Z = z$ for $z \in \{0, 1\}$, and $\tilde{f}^{(m)'}$ is an independent copy of $\tilde{f}^{(m)}$. Specifically, when $m = 0$, we have $\mathrm{dCov}^2(\tilde{f}^{(m)}, Z) = 0$ and thereby $\tilde{f}^{(m)} \perp\!\!\!\perp Z$.*

This expression reveals that distance covariance is determined by the discrepancy between within-group and between-group variations in predictions. When the reduced representation $\boldsymbol{\Xi}^{(m)}$ sufficiently removes dependence on $Z$, the term $\mathbb{E}[|\tilde{f}_1^{(m)} - \tilde{f}_0^{(m)}|]$ approaches the population-level variation $\mathbb{E}[|\tilde{f}^{(m)} - \tilde{f}^{(m)'}|]$, leading to a smaller dCov and improved fairness.

# 4 Subspace Estimation and Algorithm Implementation

## 4.1 Estimation of Projections

We describe the procedure to estimate the sufficient directions for predicting $\boldsymbol{Y}$. The first step is to estimate the intersection subspace $\mathcal{S}_{\boldsymbol{Y}|\boldsymbol{X}} \cap \mathcal{S}_{\boldsymbol{Z}|\boldsymbol{X}}$, spanned by $\{\phi_1, \ldots, \phi_s\}$. This is equivalent to estimating a matrix $\Phi \in \mathbb{R}^{p \times s}$ such that

$$\boldsymbol{Y} \perp\!\!\!\perp \boldsymbol{Z} \mid \Phi^\top \boldsymbol{X}, \tag{6}$$

which corresponds to dimension reduction with respect to the interaction between response variables as proposed in Luo [2022].

Let $\Sigma$ denote the covariance matrix of $\boldsymbol{X}$. Note that $M_Y$ and $M_Z$ are the symmetric candidate matrices from SDR methods that satisfy (4), with estimates $\widehat{M}_Y$ and $\widehat{M}_Z$. Define the cross-matrix $M_{Y,Z} = M_Y \Sigma M_Z$, and let $s = \mathrm{rank}(M_{Y,Z})$. Luo [2022] show that $\Phi$ satisfies (6) if and only if

$$M_Y \Sigma P_{\Sigma,\Phi} M_Z = M_{Y,Z}, \tag{7}$$

where $P_{\Sigma,B} = B(B^\top \Sigma B)^{-1} B^\top \Sigma$. The following theorem characterize the estimation of intersection subspace as a generalized eigenvalue decomposition problem.

**Theorem 4.1.** *Suppose $\mathrm{Span}\{\Phi\} \subseteq \mathbb{R}^p$ is an $s$-dimensional subspace. Then $\mathrm{Span}\{\Phi\}$ is given by the span of the leading $s$ eigenvectors of the following generalized eigenvalue decomposition problem: $M_Y \Sigma M_Z \nu = \lambda \Sigma \nu$. The candidate symmetric matrix for (6) is $M_{Y,Z} = M_Y \Sigma M_Z$.*

Then the estimation procedure of $\widehat{\Phi}$ is as follows. First, construct $\widehat{M}_Y$ and $\widehat{M}_Z$ using any exhaustive inverse regression method (e.g., SIR Li [1991], SAVE Cook and Weisberg [1991], or directional regression Li and Wang [2007]), and compute the sample covariance matrix $\widehat{\Sigma}$. Next, form $\widehat{M}_{Y,Z} = \widehat{M}_Y \widehat{\Sigma} \widehat{M}_Z$ and estimate the dimension $\hat{s} = \mathrm{rank}(\widehat{M}_{Y,Z})$ using the ladle estimator [Luo and Li, 2016].

Once $\widehat{\Phi}$ is estimated, the remaining directions relevant for predicting $\boldsymbol{Y}$ can be obtained via projection. Rather than estimating $\beta_1, \ldots, \beta_{q-s}$ directly, we estimate their projections onto the orthogonal complement of $\mathcal{S}_{\boldsymbol{Z}|\boldsymbol{X}}$. Define the estimated projection matrix $\widehat{Q}_z = I_p - \sum_{j=1}^{\hat{r}} \hat{\psi}_j \hat{\psi}_j^\top$, then we apply any SDR method to $(\boldsymbol{Y}, \widehat{Q}_z \boldsymbol{X})$ to obtain $\widehat{B}_{Y,Q_z}$ such that $\boldsymbol{Y} \perp\!\!\!\perp \widehat{P}_z \boldsymbol{X} \mid \widehat{B}_{Y,Q_z}^\top \boldsymbol{X}$. This matrix approximates the projected directions $\beta_1^\top P_z, \ldots, \beta_{q-s}^\top P_z$. Together, the estimated shared directions $\widehat{\Phi}$ and the unshared components $\widehat{B}_{Y,P_z}$ form a sufficient projection $\widehat{P}^{(m)}$. The consistency and $n^{-1/2}$ convergence rates of the estimated directions and projections are well-established in the SDR literature; we omit these details for brevity. The complete procedure is summarized in Algorithm 1.

*Remark* 4.2. The computational cost of obtaining the fair projection matrix primarily arises from estimating the candidate matrices $M_Y$, $M_Z$ and $M_{Y,Z}$, each typically constructed as a weighted covariance matrix. These computations scale linearly with the sample size $n$. In addition, the procedure includes an eigen-decomposition step for $p \times p$ matrices, and the rank estimation step scales linearly with dimension $p$.

---

**Algorithm 1** Estimation of Sufficient Projections

---

1: **Input:** Data $(X_i, Y_i, Z_i)_{i=1}^n$; SDR method for candidate matrix construction
2: **Output:** Estimated sufficient projections $\widehat{P}^{(m)}$, $m = 0, \ldots, \hat{s}$.
3: Compute $\widehat{M}_Y$, $\widehat{M}_Z$ using SDR method; compute $\widehat{\Sigma} = \mathrm{cov}(X)$.
4: Form $\widehat{M}_{Y,Z} = \widehat{M}_Y \widehat{\Sigma} \widehat{M}_Z$; apply ladle estimator to estimate rank $\hat{s}$.
5: Obtain estimator $\widehat{\Phi} \in \mathbb{R}^{p \times \hat{s}}$ and get projection $\widehat{Q}_z$.
6: Apply SDR to $(Y, \widehat{Q}_z X)$ and obtain $\widehat{B}_{Y,Q_z}$ with rank $\hat{d}_{Y,Q_z}$.
7: Obtain $\widehat{P}^{(m)} = \widehat{B}_{Y,P_z} \widehat{B}_{Y,P_z}^\top + \widehat{\Phi}_m \widehat{\Phi}_m^\top$ for $m = 0, \ldots, \hat{s}$

---

## 4.2 Algorithm Implementation

The sufficient variables $\boldsymbol{\Xi}^{(m)} = \widehat{P}^{(m)} \boldsymbol{X}$ can then be used to fit regression or classification models for downstream tasks. Note that the columns of $\widehat{\Phi}$ are ordered by the eigenvalues of $\widehat{M}_{Y,Z}$, reflect their predictive power for both $\boldsymbol{Y}$ and $\boldsymbol{Z}$. By gradually adding these columns in the projection, we can incrementally build models with varying levels of sensitive information included.

This setup allows users to manually control the amount of sensitive information retained when predicting $\boldsymbol{Y}$. The post-SDR training procedure is summarized in Algorithm 2, using a classification task with accuracy (Acc) as the utility metric and demographic parity (DP) as the fairness metric. The optimal feature set and fair model are chosen to achieve at least 95% of the validation accuracy from the full dataset while minimizing unfairness.

*Remark* 4.3. Algorithm 2 presents one example of post-SDR training. In practice, DP and Acc can be replaced by any fairness and utility metrics, and the 95% threshold generalized to any $\tau \in (0, 1)$. The shared dimension $s$ acts as a tuning parameter analogous to a regularization coefficient, offering two advantages: (1) $s$ is selected from a finite set, avoiding continuous grid search (one can step every 2–3 dimensions or use bisection); and (2) each added component remains interpretable, enabling clear insight into the fairness–performance trade-off.

---

**Algorithm 2** Sequential Fair Projection: Post-SDR Fair Model Training

---

1: **Input:** Sufficient variables $\boldsymbol{\Xi}^{(m)} = \widehat{P}^{(m)} \boldsymbol{X}$, training and validation datasets
2: **Output:** Fair model $\mathcal{M}_{\text{fair}}$ and selected feature set $\boldsymbol{\Xi}^{(m^*)}$
3: **for** $i = 0$ to $s$ **do**
4:     Train model $\mathcal{M}_{\text{fair}}^{(i)}$ based on $\boldsymbol{\Xi}^{(i)}$ and record $\mathrm{Acc}^{(i)}$ and $\mathrm{DP}^{(i)}$ on validation set.
5: **end for**
6: Let $s^* = \arg\min_i \{\mathrm{DP}^{(i)} \mid \mathrm{Acc}^{(i)} > 95\% \, \mathrm{Acc}^{(orig)}\}$, where $\mathrm{Acc}^{(orig)}$ is the accuracy trained by original datasets $\boldsymbol{X}$.
7: **return** Selected model $\mathcal{M}_{\text{fair}} = \mathcal{M}_{\text{fair}}^{m^*}$ and corresponding feature set $\boldsymbol{\Xi}^{(m^*)}$.

---

# 5 Theoretical Analysis

In this section, we analyze the impact of using sufficient variables in the post-SDR training procedure, with a focus on how they influence parameter estimation and expected errors. We employ influence functions to trace a model's prediction through the learning algorithm and back to its input features.

## 5.1 Influence Functions

Let $(\boldsymbol{X}, \boldsymbol{Y}, \boldsymbol{Z}) \sim F_0$ be the joint law and we denote the empirical distribution on $n$ samples of $(\boldsymbol{X}, \boldsymbol{Y}, \boldsymbol{Z})$ as $F_n$. Define the SDR functional $M_Y(F)$ and $M_Z(F)$ for estimating central subspaces $\mathcal{S}_{Y|X}$ and $\mathcal{S}_{Z|X}$ respectively, and let $\Sigma(F) = \mathrm{Var}(X)$ be the functional for the covariance matrix. Therefore, the intersection subspace estimation for $\Phi$ corresponds to reduction functional $M_{Y,Z}(F) = M_Y(F)\Sigma(F)M_Z(F)$.

In the following, an asterisk on a symbol always indicates the influence function of a statistical functional represented by that symbol. For a sample point $S = (x, y, z)$ drawn from $F_0$, let $\delta_S$ be the

Dirac measure at $S$. The influence function of the functional $R$ is defined as

$$R^*(S) = \frac{\partial}{\partial \varepsilon} R[(1-\varepsilon)F_0 + \varepsilon\delta_S] \mid_{\varepsilon=0}.$$

For notation simplicity, we abbreviate $R^*(S)$ by $R^*$. In the following, an asterisk on a symbol always indicates the influence function of a statistical functional represented by that symbol. For example, we denote the influence functions of functionals $M_Y(F)$, $M_Z(F)$, $M_{Y,Z}(F)$ and $\Sigma(F)$ as $M_Y^*$, $M_Z^*$, $M_{Y,Z}^*$ and $\Sigma^*$, respectively. For notation simplicity, we omit the Using the product rule for Gateaux derivatives, we obtain the influence function of $M_{Y,Z}^*$.

**Lemma 5.1.** *Suppose $M_Y(F)$, $M_Z(F)$ and $\Sigma(F)$ are Hadamard differentiable. Then, the influence function of the reduction functional $M_{Y,Z}(F)$ is*

$$M_{Y,Z}^* = M_Y^* \Sigma(F) M_Z(F) + M_Y(F) \Sigma^* M_Z(F) + M_Y(F) \Sigma(F) M_Z^*.$$

## 5.2 Asymptotic Normality of Estimators

Let $f(X;\theta)$ be the differentiable predictive function parametrized by $\theta \in \Theta$ and let $L(x, y; \theta)$ be the differentiable loss function with respect to $\theta$, where we fold in any regularization terms into $L$. For notation simplicity, we denote $L(x, y; \theta)$ by $L(S; \theta)$. Then the population and empirical risk minimizer of the parameter is given by

$$\widetilde{\theta} = \theta(F_0) \triangleq \arg\min_{\theta \in \Theta} \mathbb{E}[L(S, \theta)], \quad \widehat{\theta}_n = \theta(F_n) \triangleq \arg\min_{\theta \in \Theta} \frac{1}{n} \sum_{i=1}^n L(S_i, \theta)$$

Similarly, we define the parameters estimated by sufficient variables $S^{(m)} = (P^{(m)}x, y, z)$ as

$$\widetilde{\theta}^{(m)} = \theta^{(m)}(F_0, P^{(m)}(F_0)) \triangleq \arg\min_{\theta \in \Theta} \mathbb{E}[L(S^{(m)}, \theta)]$$

$$\widehat{\theta}_n^{(m)} = \theta^{(m)}(F_n, P^{(m)}(F_n)) \triangleq \arg\min_{\theta \in \Theta} \frac{1}{n} \sum_{i=1}^n L(\widehat{S}_i^{(m)}, \theta),$$

where $P^{(m)}(F_n) = \widehat{P}^{(m)}$ and $\widehat{S}^{(m)} = (\widehat{P}^{(m)}x, y, z)$. We refer to $\widehat{\theta}_n$ as the original estimator and $\widehat{\theta}_n^{(m)}$ as the fair estimator. The entire post-SDR learning procedure can be viewed as a composition of three mappings:

$$F_n \xrightarrow{\text{reduction}} P^{(m)}(F_n) \xrightarrow{\text{estimation}} \theta^{(m)}(F_n, P^{(m)}(F_n)) \xrightarrow{\text{prediction}} f(\cdot; \theta^{(m)}(F_n, P^{(m)}(F_n))).$$

The composition mapping allows us to derive the asymptotic normality of both the estimators and predictors after applying fair projections, as stated in the following theorems.

**Theorem 5.2.** *Suppose all the statistical functionals presented above are Hadamard differentiable, and their influence functions has zero expectation and finite variance, then we have the following asymptotic normality for the estimator $\widehat{\theta}_n^{(m)}$*

$$\sqrt{n}(\widehat{\theta}_n^{(m)} - \widetilde{\theta}^{(m)}) \xrightarrow{\mathcal{D}} \mathcal{N}\left(0, \mathrm{Var}\left[-H_{\widetilde{\theta}^{(m)}}^{-1} G^{(m)} + D^{(m)} \mathrm{vec}(P^{(m)*})\right]\right), \tag{8}$$

*where $H_{\widetilde{\theta}^{(m)}}^{-1} = \mathbb{E}[\nabla_\theta^2 L(S^{(m)}, \widetilde{\theta}^{(m)})]$ is the expected Hessian of loss function, $G^{(m)} = \nabla_\theta L(S^{(m)}, \widetilde{\theta}^{(m)})$ is the gradient, $D^{(m)} = (\partial\theta(F_0, P^{(m)}(F_0))/\partial\mathrm{vec}(P^{(m)}))^\top$, $\mathrm{vec}(\cdot)$ is vectorization of the matrix, and $P^{(m)*}$ denotes the influence function of $P^{(m)}(F)$, whose explicit form depends on the choice of SDR method and is provided in the Appendix.*

**Corollary 5.3.** *Under the same assumptions stated in Theorem 5.2, we have*

$$\mathbb{E}(\|\widehat{\theta}_n^{(m)} - \widehat{\theta}_n\|^2) \le \|\widetilde{\theta}^{(m)} - \widetilde{\theta}\|^2 + \frac{1}{n} \mathrm{Tr}\left(\mathrm{Var}\left[H_{\widetilde{\theta}}^{-1} G - H_{\widetilde{\theta}^{(m)}}^{-1} G^{(m)} + D^{(m)} \mathrm{vec}(P^{(m)*})\right]\right), \tag{9}$$

*where $H_{\widetilde{\theta}}^{-1} = \mathbb{E}[\nabla_\theta^2 L(S, \widetilde{\theta})]$ and $G = \nabla_\theta L(S, \widetilde{\theta})$.*

**Theorem 5.4.** *Let $f(x; \theta)$ be the predictor evaluated at covariate $x$. Under the same assumptions stated in Theorem 5.2, we have the following asymptotic normality for the predictor*

$$\sqrt{n}\left(f(x; \widehat{\theta}_n^{(m)}) - f(x; \widetilde{\theta}^{(m)})\right) \xrightarrow{\mathcal{D}} \mathcal{N}\left(0, g^\top \mathrm{Var}\left[-H_{\widetilde{\theta}^{(m)}}^{-1} G^{(m)} + D^{(m)} \mathrm{vec}(P^{(m)*})\right]g\right), \tag{10}$$

*where $g = \nabla_\theta f(x; \widetilde{\theta}^{(m)}))$.*

Theorems 5.2 and 5.4 provide feasible inference procedures for the fair estimators corresponding to each $m$, which are useful for subsequent statistical inference or hypothesis testing. Corollary 5.3 follows directly from Theorem 5.2 by comparing the asymptotic distributions of the fair and original estimators. It shows that the expected distance between the fair and original estimators is upper bounded by the distance between their respective true parameters and the sampling variability.

Moreover, the projection matrix $P^{(m)}$ plays a crucial role in shaping the asymptotic variance of the fair estimator. By restricting the estimation to a subspace spanned by $P^{(m)}$, it reduces the dimensionality of the parameter space, leading to smaller asymptotic variance. However, smaller $m$ also implies that important predictive directions are truncated and inflate the $\|\widetilde{\theta}^{(m)} - \widetilde{\theta}\|^2$ term. Thus, $P^{(m)}$ provides a natural mechanism to balance variance reduction and bias introduction.

# 6 Experiments

## 6.1 Simulation Studies

In this section, we use simulation results to justify the behavior of using the sufficient variables under the fairness setting. Consider the multivariate linear regression, we simulate a dataset $(\boldsymbol{X}, \boldsymbol{Y}, \boldsymbol{Z})$, where the sensitive attribute $\boldsymbol{Z} \in \{0, 1\}$ is binary.

Let $A$ be a randomly generated matrix with columns are orthonormal. Denote $A_Y \in \mathbb{R}^{p \times q}$ is the first $q$ columns of $A$ and $A_Z \in \mathbb{R}^{p \times r}$ is the $q - s$ to $q - s + r$ columns of A, which means $A_Y$ and $A_Z$ has $s$ shared columns. Define the latent variables $\boldsymbol{U}_Y = \boldsymbol{X} A_Y A_Y^\top$ and $\boldsymbol{U}_Z = \boldsymbol{X} A_Z A_Z^\top$. Let $\boldsymbol{X} \in \mathbb{R}^p$ drawn from a standard multivariate normal distribution: $\boldsymbol{X} \sim \mathcal{N}(\mathbf{0}, I_p)$. The response $\boldsymbol{Y} \in \mathbb{R}^K$ is generated from

$$\boldsymbol{Y} = \boldsymbol{U}_Y \boldsymbol{\theta} + \varepsilon_Y, \quad \varepsilon_Y \sim \mathcal{N}(0, 0.5^2 I_K),$$

where the coefficients $\boldsymbol{\theta} \in \mathbb{R}^{p \times K}$ are randomly generated with each entry samples from $\mathcal{N}(1, 1)$. The sensitive variables $\boldsymbol{Z}$ is generated based on the following latent score

$$\boldsymbol{Z} = 1 \text{ if } \xi \geq 1, \text{ and } \boldsymbol{Z} = 0 \text{ otherwise}; \quad \text{where } \xi = \frac{1}{p} \sum_{j=1}^{p} \tanh([\boldsymbol{U}_Z]_j) + \varepsilon_Z \text{ and } \varepsilon_Z \sim \mathcal{N}(0, 1).$$

Finally, we introduce a distributional shift between groups by applying a non-linear transformation to the $\boldsymbol{X}$ samples based only on the span of $A_Z$. Specifically, for all samples $\boldsymbol{X}$ with $\boldsymbol{Z} = 1$,

$$\boldsymbol{X} \leftarrow \boldsymbol{X} + 0.5 \cdot \exp\left(\boldsymbol{X} A_Z A_Z^\top\right).$$

We generate synthetic data with parameters $n = 5000$, $p = 10$, $K = 5$, $q = 8$, $r = 8$, and $s = 6$. The dataset is randomly split into 4000 training samples and 1000 testing samples. We fit a multivariate ordinary least squares (OLS) model on both the original features $\boldsymbol{X}$ and the projected representations $P^{(m)} \boldsymbol{X}$ to estimate the model parameters $\widehat{\boldsymbol{\theta}}$.

To assess performance, we repeat the entire process over 30 independent replications. For each trial, we compute the root mean squared error (RMSE) on the test set overall and separately for the subgroups $\boldsymbol{Z} = 0$ and $\boldsymbol{Z} = 1$, along with the RMSE gap between groups. We also report the parameter distance $\|\widehat{\boldsymbol{\theta}}_n^{(m)} - \widehat{\boldsymbol{\theta}}_n\|$, where $\widehat{\boldsymbol{\theta}}_n^{(m)}$ is the OLS estimator obtained from the projected data $P^{(m)} \boldsymbol{X}$, and $\widehat{\boldsymbol{\theta}}_n$ is the baseline estimator from the original data. Results are averaged over the 30 trials and summarized in the left panel of Figure 1. We also project $\boldsymbol{X}$ onto the direction that best discriminates the sensitive attribute $\boldsymbol{Z}$ using Linear Discriminant Analysis (LDA), in order to visualize the distributional discrepancy between the two sensitive groups.

The simulation results clearly support our theoretical findings. Specifically, as the shared dimension increases, more directions in $\Phi$ are used to predict the target variable, incorporating additional sensitive and predictive information. This leads to a reduction in the MSE of the predictor and the parameter distance to the original estimator, but an increase in the MSE gap between sensitive groups. Both the distribution discrepancy and the Wasserstein distance also increase, indicating growing divergence between groups as more sensitive information is included.

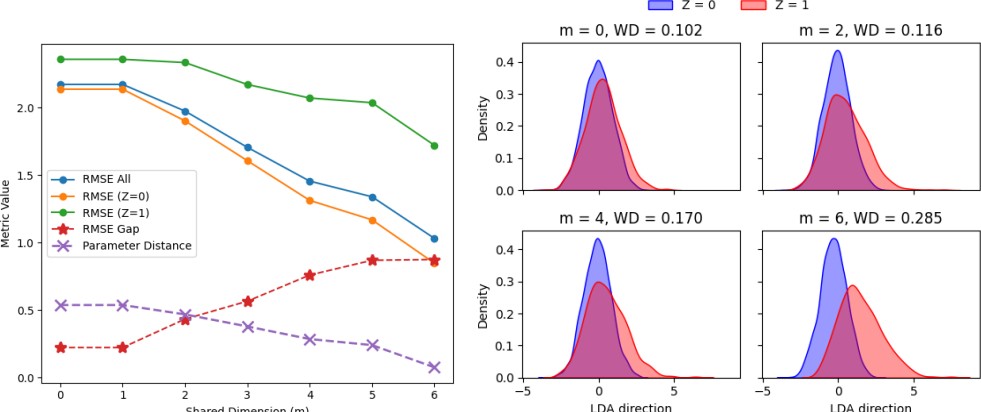

Figure 1: Left panel: Trends of RMSE and parameter distance as the number of shared dimensions increases, averaged over 30 replications. Right panel: Distributional discrepancy between sensitive groups visualized via LDA, along with the average Wasserstein distance (WD) across all $p$ dimensions between the original and projected data as the shared dimension increases in one replication.

## 6.2 Real Data Applications

We evaluate our proposed **Sequential Fair Projection (SFP)** method on two tabular datasets: **Adult** Kohavi [1996] and **Bank** Moro et al. [2014]. The **Adult** dataset contains personal data from over 40K individuals, with the task of predicting whether annual income exceeds $50K; gender is used as the sensitive attribute. The **Bank** dataset originates from Portuguese marketing campaigns, where the goal is to predict whether a client will subscribe to a deposit; age (over/under 25) is treated as the sensitive attribute. The data is standardized during preprocessing and split into training, validation, and test sets with a ratio of 70%:10%:20%.

We compare **SFP** against the following baselines: **Logistic Regression (LR)**, **AdvDebias** Zhang et al. [2018], **FairMixup** Chuang and Mroueh [2021], **DRAlign** Li et al. [2023], **DiffMCDP** Jin et al. [2024], **INLP** Ravfogel et al. [2020], **RLACE** Ravfogel et al. [2022], and **SUP** Shi et al. [2024]. Fairness is evaluated using TPR gap, DP, and MCDP, and utility is evaluated using accuracy. The detailed experimental settings are shown in Appendix B. All methods are repeated 20 times, and we report the average performance with standard deviations. The results are presented in Table 1.

Table 1: Performance metrics on the Adult and Bank datasets over 20 replications. Optimal results are in **bold**, and sub-optimal results are underlined.

| | **Adult Dataset** | | | | | **Bank Dataset** | | | |
|---|---|---|---|---|---|---|---|---|---|
| **Method** | **Accuracy ↑** | **DP ↓** | **TPR ↓** | **MCDP ↓** | **Method** | **Accuracy ↑** | **DP ↓** | **TPR ↓** | **MCDP ↓** |
| LR | 84.90 | 17.37 | 6.87 | 35.12 | LR | 91.17 | 6.72 | 2.62 | 26.28 |
| AdvDebias | 76.49 (0.65) | 12.45 (1.56) | 5.27 (0.53) | 29.32 (2.31) | AdvDebias | 60.56 (1.00) | 2.01 (1.36) | 2.20 (0.13) | 17.55 (1.39) |
| FairMixup | 74.48 (0.43) | 3.93 (1.34) | 5.11 (0.34) | 24.91 (1.58) | FairMixup | 59.48 (1.96) | 1.07 (0.25) | 2.21 (0.24) | 13.18 (2.94) |
| DRAlign | 75.01 (0.41) | 7.58 (1.03) | 4.78 (0.29) | 22.04 (1.22) | DRAlign | 59.12 (1.56) | 1.16 (0.41) | **1.96 (0.21)** | 13.61 (1.23) |
| DiffMCDP | 73.93 (0.31) | 5.92 (1.25) | 5.33 (0.46) | 11.50 (1.09) | DiffMCDP | 60.01 (1.83) | 1.19 (0.39) | 2.18 (0.13) | 11.00 (0.97) |
| INLP | 68.27 (0.57) | 4.16 (0.34) | 4.79 (0.51) | 8.58 (1.24) | INLP | 70.13 (0.86) | 0.93 (0.21) | 2.38 (0.31) | 10.44 (0.63) |
| RLACE | 72.79 (0.83) | 5.24 (0.56) | 4.56 (0.31) | **7.45 (0.86)** | RLACE | 72.51 (0.53) | 1.02 (0.35) | 2.26 (0.27) | **8.98 (0.38)** |
| SUP | 70.53 (0.36) | 4.33 (0.27) | **4.37 (0.29)** | 8.16 (1.25) | SUP | 70.82 (0.47) | 0.96 (0.16) | 2.51 (0.22) | 10.63 (0.42) |
| **SFP (Ours)** | **76.88 (0.47)** | **3.78 (0.15)** | 4.43 (0.26) | 10.52 (0.72) | **SFP (Ours)** | **89.66 (0.27)** | **0.51 (0.26)** | 2.33 (0.27) | 10.57 (0.34) |

Our results show that SFP consistently strikes an effective balance between predictive accuracy and fairness. On both the Adult and Bank datasets, SFP achieves competitive or superior accuracy compared to existing fairness-aware methods while substantially reducing group-level disparities. On the Adult dataset, SFP attains the highest accuracy (76.88%) and the lowest DP gap, demonstrating strong control over statistical bias. It also performs well in terms of TPR gap and MCDP, with only minor trade-offs relative to sub-optimal methods. On the Bank dataset, SFP also achieves the highest accuracy (89.66%) and the lowest DP gap (0.51), indicating minimal disparity. Although RLACE slightly outperforms in MCDP, SFP maintains competitive fairness across all metrics. These findings

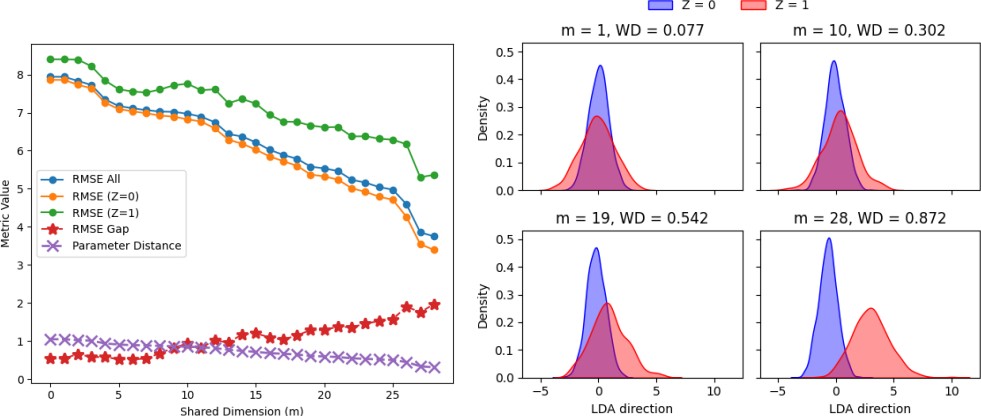

Figure 2: Trends of RMSE, parameter distance and distributional discrepancy as the number of shared dimensions increases when the linear SDR assumption is violated.

confirm that SFP is a practical and flexible framework for fairness-aware learning, enabling users to suppress sensitive information while preserving task-relevant predictive power.

### 6.3 Model Misspecification

In practice, the linear SDR assumption could be violated and the estimated central subspaces will fail to capture the full conditional independence structure. Therefore, conditional independence tests can be applied to verify whether the estimated projections preserve sufficiency. When the linear SDR assumption is strongly violated, the estimated matrices $M_Y$ and $M_Z$ may lose their low-rank structure. This indicates that no low-dimensional linear subspace can capture the dependence between $(X, Y)$ or $(X, Z)$. In such cases, one practical remedy is to retain a subset of directions corresponding to the leading eigenvalues and $M_{Y,Z}$ and construct a fair projection by eliminating leading directions.

To examine the trade-off between utility and fairness of SFP under model misspecification, we conduct additional experiments based on the simulation setup described in Section 6.1, with an added nonlinear term $\|X\|_2^2/p$ in the generating processes of both $Y$ and $\xi$. This modification violates the linear SDR assumption. We set $p = 30$, $K = 5$, and $q = r = s = 30$, making it impossible to recover a low-rank representation through SDR. After applying SFP, the estimated shared dimension is $\hat{s} = 28$, indicating that the rank of $\widehat{M}_{Y,Z}$ is 28. We then sequentially add the directions obtained from $\widehat{M}_{Y,Z}$ and report the resulting trend in Figure 2.

The overall RMSE and RMSE gap using the original representation are 3.18 and 3.06, respectively. As shown in Figure 2, there is a clear trend that, as the number of shared dimensions used to construct the projection matrix increases, both the overall MSE and the parameter distance decrease, indicating that the projected representation gradually approaches the information contained in the original features. Meanwhile, the RMSE gap and the distributional discrepancy increase as more sensitive information is included. This demonstrates that SFP can still capture the trade-off between utility and fairness even when the linear SDR assumption is violated.

## 7 Discussion

We propose a principled and model-agnostic framework for fairness-aware learning through subspace decomposition of data representations. Our method manipulates the representation space by selectively removing shared information between the target and sensitive attributes, which enables flexible control over the fairness-utility trade-off. We provide theoretical guarantees on how prediction error and fairness metrics evolve as more sensitive information is incorporated, and further apply influence function analysis to characterize the impact of partially fair representations on estimator behavior. Empirical results on both synthetic and real-world datasets validate our theoretical insights, demonstrating that the proposed SFP method achieves good performance.

## Acknowledgements

Bei Jiang and Linglong Kong were partially supported by grants from the Canada CIFAR AI Chairs program, the Alberta Machine Intelligence Institute (AMII), and Natural Sciences and Engineering Council of Canada (NSERC), and Linglong Kong was also partially supported by grants from the Canada Research Chair program from NSERC.

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

# A Influence Function of Projections

In this section, we give explicit form of the influence function of projection $P^{(m)}$, which can used to derive the asymptotic normality of the parameter estimators.

Let $\{(\lambda_{\phi,i}, \phi_i)\}_{i=1}^s$ be the eigenvalues and associated eigenvectors of $M_{Y,Z}$, $\{(\lambda_{\beta,i}, \beta_i)\}_{i=1}^q$ be the eigenvalues and associated eigenvectors of $M_Y$ and $\{(\lambda_{\psi,i}, \psi_i)\}_{i=1}^r$ be the eigenvalues and associated eigenvectors of $M_Z$.

Then, by Zhu and Fang [1996], the influence function of the directions estimated by SDR techniques can be written as, for $i = 1, \ldots, s$,

$$\phi_i^* = \sum_{j=1,\, j \neq i}^s \frac{\phi_j \phi_j^\top M_{Y,Z}^* \phi_i}{\lambda_{\phi,i} - \lambda_{\phi,j}} \tag{11}$$

And for $k = 1, \ldots, q - s$

$$(Q_z \beta_k)^* = Q_z \beta_k^* + Q_z^* \beta_k = Q_z \beta_k^* + \beta_k \left( I_p - \sum_{j=1}^r (\psi_j^* \psi_j^\top + \psi_j \psi_j^{*\top}) \right) \tag{12}$$

where

$$\beta_k^* = \sum_{\ell=1,\, \ell \neq k}^q \frac{\beta_\ell \beta_\ell^\top M_Y^* \beta_k}{\lambda_{\beta,k} - \lambda_{\beta,\ell}} \quad \text{and} \quad \psi_j^* = \sum_{\ell=1,\, \ell \neq j}^r \frac{\psi_\ell \psi_\ell^\top M_Z^* \psi_j}{\lambda_{\psi,j} - \lambda_{\psi,\ell}}.$$

Then we have

$$P^{(m)*} = (Q_z \beta_k)^* (Q_z \beta_k)^\top + (Q_z \beta_k)(Q_z \beta_k)^{*\top} + \Phi_m^* \Phi_m^\top + \Phi_m \Phi_m^{*\top},$$

where $\Phi_m^* = (\phi_1^*, \ldots, \phi_m^*)$.

The influence functions for the candidate matrices $M_Y$ and $M_Z$ estimated via SDR techniques have been well studied. For details, we refer readers to Section 4 of Kim et al. [2020] and omit the derivations here.

# B Experiments Details

## B.1 Experiment Setup

Throughout the experiments, we use the MSAVE method to estimate the candidate matrices $M_Y$ and $M_Z$, with the number of slices set to $p + 1$, where $p$ denotes the dimensionality of the input $\boldsymbol{X}$. In the post-SDR training procedure (as described in Algorithm 2), we apply multivariate linear regression for synthetic simulations and logistic regression for real-world datasets. The MCDP metric is used to select the optimal fair model $\mathcal{M}_{\text{fair}}^{(m^*)}$. To estimate the rank of each candidate matrix, we adopt the ladle estimator with 30 bootstrap replications.

For baseline methods, we use the official code released by the respective authors. Specifically, for AdvDebias, FairMixup, DRAlign, and DiffMCDP, we use the implementation provided in Jin et al. [2024]. For INLP, we set the number of iterations to 100. For RLACE, we follow the training hyperparameters used in the original paper. For SUP, we adopt the same number of slices ($p + 1$) as in our method, and select the dimension to remove based on 10-fold cross-validation, choosing the model that achieves the lowest MCDP.

## B.2 Fairness Measurement

We consider a multi-class classification setting where the target label $\boldsymbol{Y} \in \mathbb{R}^K$ is a one-hot encoded vector, i.e., $\boldsymbol{Y}_j = 1$ if the true label corresponds to class $j$ and 0 otherwise. We denote the predicted probability vector as $\hat{\boldsymbol{Y}}$, and the sensitive attribute as a binary variable $\boldsymbol{Z} \in \{0, 1\}$.

**Demographic Parity (DP) Gap.** Demographic Parity requires that the predicted output is independent of the sensitive attribute. For each class $j \in \{1, \ldots, K\}$, we define the group-wise expected predicted score as:

$$\text{DP}_{z,j} = \mathbb{E}[\hat{\boldsymbol{Y}}_j \mid \boldsymbol{Z} = z],$$

and the corresponding DP gap for class $j$ as:

$$\text{DP}_{\text{gap},j} = \text{DP}_{1,j} - \text{DP}_{0,j}.$$

The overall DP gap across classes is then aggregated as:

$$\text{DP}_{\text{gap}} = \sqrt{\frac{1}{K-1} \sum_{j=1}^{K-1} (\text{DP}_{\text{gap},j})^2} \times 100\%.$$

When $K = 2$, this definition reduces to the binary case, where $\text{DP}_{\text{gap}} = |\text{DP}_{1,j} - \text{DP}_{0,j}|$.

**True Positive Rate (TPR) Gap.** For each class $j \in \{1, \ldots, K\}$, the TPR for group $z \in \{0, 1\}$ is defined as:

$$\text{TPR}_{z,j} = \mathbb{P}(\hat{\boldsymbol{Y}}_j = 1 \mid \boldsymbol{Z} = z, \boldsymbol{Y}_j = 1),$$

and the corresponding TPR gap is:

$$\text{TPR}_{\text{gap},j} = \text{TPR}_{1,j} - \text{TPR}_{0,j}.$$

We aggregate TPR gaps across all classes into a single fairness metric:

$$\text{TPR}_{\text{gap}} = \sqrt{\frac{1}{K} \sum_{j=1}^{K} (\text{TPR}_{\text{gap},j})^2} \times 100\%.$$

**Maximal Cumulative Ratio Disparity along Predictions (MCDP).** For each class $j$, define the group-wise cumulative distribution function:

$$F_{z,j}(y) = \mathbb{P}(\hat{\boldsymbol{Y}}_j \leq y \mid \boldsymbol{Z} = z).$$

The MCDP for class $j$ is then the Kolmogorov–Smirnov distance between the distributions of predicted probabilities for the two sensitive groups:

$$\text{MCDP}_j = \max_{y \in [0,1]} |F_{1,j}(y) - F_{0,j}(y)|.$$

Since the components of $\hat{\boldsymbol{Y}}$ sum to 1 due to the softmax constraint, only $K - 1$ of them are linearly independent. Therefore, we define the overall MCDP gap as:

$$\text{MCDP}_{\text{gap}} = \sqrt{\frac{1}{K-1} \sum_{j=1}^{K-1} (\text{MCDP}_j)^2} \times 100\%.$$

When $K = 2$, this definition reduces to the binary case as in Jin et al. [2024].

## C   Proof of Theorems

*Proof of Theorem 3.1.* Let $B$ be an orthonormal basis for the central subspace $\mathcal{S}_{\boldsymbol{Y}|\boldsymbol{X}}$, so that $\mathbb{E}[\boldsymbol{Y} \mid \boldsymbol{X}] = \mathbb{E}[\boldsymbol{Y} \mid B^\top \boldsymbol{X}]$. Since $Q_z \boldsymbol{X}$ is a measurable function of $\boldsymbol{X}$, and $\boldsymbol{Y} \perp \boldsymbol{X} \mid B^\top \boldsymbol{X}$, it follows that $\mathbb{E}[\boldsymbol{Y} \mid Q_z \boldsymbol{X}] = \mathbb{E}[\boldsymbol{Y} \mid B^\top Q_z \boldsymbol{X}]$. Noting that $B^\top Q_z \boldsymbol{X} = (Q_z B)^\top \boldsymbol{X}$, we conclude that

$$\mathbb{E}[\boldsymbol{Y} \mid Q_z \boldsymbol{X}] = \mathbb{E}[\boldsymbol{Y} \mid (Q_z B)^\top \boldsymbol{X}],$$

which implies that $Q_z B$ spans a sufficient subspace for $\boldsymbol{Y}$ with respect to $Q_z \boldsymbol{X}$.

By definition, the central subspace for $\boldsymbol{Y} \mid Q_z \boldsymbol{X}$ is the minimal subspace satisfying this conditional independence. Therefore, if $\widetilde{B}$ is any orthonormal matrix such that $\mathbb{E}[\boldsymbol{Y} \mid Q_z \boldsymbol{X}] = \mathbb{E}[\boldsymbol{Y} \mid \widetilde{B}^\top \boldsymbol{X}]$, it must hold that

$$\text{Span}(\widetilde{B}) \subseteq \text{Span}(Q_z B).$$

and $\text{Span}(\widetilde{B}) = \text{Span}(Q_z B)$ if and only if $\text{rank}(\widetilde{B}) = \text{rank}(Q_z B)$ $\qquad \square$

*Proof of Theorem 3.2.* We begin by substituting the model into the expected error:

$$\mathbb{E}[(Y - \tilde{f}^{(m)})^2] = \mathbb{E}[(f^*(\boldsymbol{X}) + \varepsilon_Y - \tilde{f}^{(m)})^2] = \mathbb{E}\left[(f^*(\boldsymbol{X}) - \tilde{f}^{(m)})^2 + 2(f^*(\boldsymbol{X}) - \tilde{f}^{(m)})\varepsilon_Y + \varepsilon_Y^2\right].$$

Take expectations term-by-term. For the first term, we observe that $\tilde{f}^{(m)} = \mathbb{E}[Y \mid \boldsymbol{\Xi}^{(m)}]$, which is the projection of $Y$ onto a coarser sigma-algebra. Therefore $\tilde{f}^{(m)} = \mathbb{E}[f^*(\boldsymbol{X}) \mid \boldsymbol{\Xi}^{(m)}]$ and

$$\mathbb{E}[(f^*(\boldsymbol{X}) - \tilde{f}^{(m)})^2] = \mathbb{E}[\text{Var}(f^*(\boldsymbol{X}) \mid \boldsymbol{\Xi}^{(m)})].$$

For the second term, since $\mathbb{E}[\varepsilon_Y \mid \boldsymbol{\Xi}^{(m)}] = \mathbb{E}[\mathbb{E}[\varepsilon_Y \mid \boldsymbol{X}] \mid \boldsymbol{\Xi}^{(m)}] = 0$, we have:

$$\mathbb{E}[(f^*(\boldsymbol{X}) - \tilde{f}^{(m)})\varepsilon_Y] = 0.$$

Finally, the third term is simply $\mathbb{E}[\varepsilon_Y^2] = \sigma_Y^2$. Putting everything together gives:

$$\mathbb{E}[(Y - \tilde{f}^{(m)})^2] = \mathbb{E}[\text{Var}(f^*(\boldsymbol{X}) \mid \boldsymbol{\Xi}^{(m)})] + \sigma_Y^2.$$

Note that $\tilde{f}^{(m)}(\boldsymbol{\Xi}^{(m)})$ is the orthogonal projection of $f^*(\boldsymbol{X}) := \mathbb{E}[Y \mid \boldsymbol{X}]$ onto $\sigma(\boldsymbol{\Xi}^{(m)})$ in $L^2$. Since $\boldsymbol{\Xi}^{(m)} \subset \boldsymbol{\Xi}^{(m+1)}$, the projection becomes finer, and by the Pythagorean identity in $L^2$:

$$\|f^*(\boldsymbol{X}) - \tilde{f}^{(m+1)}\|_{L^2}^2 \leq \|f^*(\boldsymbol{X}) - \tilde{f}^{(m)}\|_{L^2}^2.$$

Hence, the approximation error $\Delta(m) := \mathbb{E}[\text{Var}(f^*(\boldsymbol{X}) \mid \boldsymbol{\Xi}^{(m)})]$ satisfies $\Delta(m+1) \leq \Delta(m)$.  $\square$

*Proof of Theorem 3.3.* From the definition of distance covariance for scalar $X$ and binary $Z$, we consider the V-statistic form:

$$\text{dCov}^2(X, Z) = \mathbb{E}[|X - X'| \cdot |Z - Z'|] + \mathbb{E}[|X - X'|] \cdot \mathbb{E}[|Z - Z'|] - 2\mathbb{E}[|X - X'| \cdot |Z - Z''|].$$

Specializing to $Z \in \{0, 1\}$, we know:

$$\mathbb{E}[|Z - Z'|] = \mathbb{E}[|Z - Z''|] = 2p(1-p),$$

and $|Z - Z'| = 1$ only when $Z \neq Z'$, meaning cross-group expectation. Then:

$$\mathbb{E}[|X - X'| \cdot |Z - Z'|] = 2p(1-p) \cdot \mathbb{E}[|X_0 - X_1|],$$
$$\mathbb{E}[|X - X'| \cdot |Z - Z''|] = 2p(1-p) \cdot \mathbb{E}[|X - X'|].$$

So:

$$\text{dCov}^2(X, Z) = 2p(1-p)\left(\mathbb{E}[|X_0 - X_1|] - \mathbb{E}[|X - X'|]\right).$$

Substituting $X = \tilde{f}^{(m)}$, we obtain the stated identity. Specifically, when $m = 0$, we have $\tilde{f}^{(m)} = \tilde{f}_0^{(m)} = \tilde{f}_1^{(m)}$ and thereby the distance covariance equals zero.  $\square$

*Proof of Theorem 4.1.* Following Luo [2022], $\text{Span}\{\Phi\}$ is the unique solution of

$$M_Y \, \Sigma \, P_{\Sigma, \Phi} \, M_Z \; = \; M_Y \, \Sigma \, M_Z, \tag{13}$$

where, for any full rank $B$, $P_{\Sigma, B}$ is the $\Sigma$-orthogonal projector onto $\text{Span}\{B\}$.

Observe that

$$P_{\Sigma, \Phi} = \Sigma^{-1/2} \, P_U \, \Sigma^{1/2}, \quad \text{where} \quad U = \Sigma^{1/2} \, \Phi, \quad P_U = U \, (U^\top U)^{-1} U^\top.$$

Substituting into (13) yields

$$M_Y \, \Sigma^{1/2} \, P_U \, \Sigma^{1/2} \, M_Z = M_Y \, \Sigma \, M_Z.$$

Left- and right-multiplying by $\Sigma^{-1/2}$ gives

$$A \, P_U \, B = E, \quad A = \Sigma^{-1/2} M_Y \, \Sigma^{1/2}, \quad B = \Sigma^{1/2} M_Z \, \Sigma^{-1/2}, \quad E = A \, B = \Sigma^{-1/2} M_Y \, \Sigma \, M_Z \, \Sigma^{-1/2}.$$

Since $\operatorname{range}(A^\top) \subseteq \operatorname{range}(U)$ and $\operatorname{range}(B) \subseteq \operatorname{range}(U)$, we have $A\,P_U = A$ and $P_U\,B = B$. It follows that

$$E\,P_U = E.$$

Because $E$ is symmetric, its nonzero-eigenvalue subspace coincides with $\operatorname{range}(U)$. Writing the spectral decomposition

$$E = V\,\Lambda\,V^\top, \quad V^\top V = I,$$

and setting $\Phi = \Sigma^{-1/2} V_{(s)}$ (the $s$ leading eigenvectors), we recover the equivalent generalized eigenproblem

$$(M_Y\,\Sigma\,M_Z)\,\phi = \lambda\,\Sigma\,\phi,$$

whose top $s$ eigenvectors $\{\phi_i\}$ span $\operatorname{Span}\{\Phi\}$. $\qquad\square$

*Proof of Theorem 5.2.* It is well-known that, if $\theta^{(m)}(F)$ is Hadamard differentiable, then it satisfies the following expansion

$$\theta^{(m)}(F_n, P^{(m)}(F_n)) = \theta^{(m)}(F_0, P^{(m)}(F_0)) + \mathbb{E}_n[\theta^{(m)*}] + o_p(n^{-1/2})$$

$$= \theta^{(m)}(F_0, P^{(m)}(F_0)) + \frac{1}{n}\sum_{i=1}^{n} \theta^{(m)*}(S_i, P^{(m)}(F_n)) + o_p(n^{-1/2})$$

By Theorem 1 and Proposition 1 in Kim et al. [2020], since $L(S, \theta)$ is differentiable, we have

$$\theta^{(m)*}(S, P^{(m)}(F_n)) = \theta^{(m)*}(S, P^{(m)}(F_0)) + \left(\frac{\partial\,\theta(F_0, P^{(m)}(F_0))}{\partial\,\operatorname{vec}(P^{(m)})}\right)^\top \operatorname{vec}(P^{(m)*}(S))$$

$$= -H_{\widetilde{\theta}^{(m)}}^{-1}\,\nabla_\theta L(S^{(m)}, \widetilde{\theta}^{(m)}) + \left(\frac{\partial\,\theta(F_0, P^{(m)}(F_0))}{\partial\,\operatorname{vec}(P^{(m)})}\right)^\top \operatorname{vec}(P^{(m)*}(S)),$$

where $H_{\widetilde{\theta}^{(m)}}^{-1} = \mathbb{E}[\nabla_\theta^2 L(S, \widetilde{\theta}^{(m)})]$. Therefore, we have

$$\widehat{\theta}_n^{(m)} = \widetilde{\theta}^{(m)} + \frac{1}{n}\sum_{i=1}^{n} \theta^{(m)*}(S_i, P^{(m)}(F_n)) + o_p(n^{-1/2})$$

$$= \widetilde{\theta}^{(m)} + \frac{1}{n}\sum_{i=1}^{n}\left[-H_{\widetilde{\theta}^{(m)}}^{-1}\,\nabla_\theta L(S_i^{(m)}, \widetilde{\theta}^{(m)}) + \left(\frac{\partial\,\theta(F_0, P^{(m)}(F_0))}{\partial\,\operatorname{vec}(P^{(m)})}\right)^\top \operatorname{vec}(P^{(m)*}(S_i))\right] + o_p(n^{-1/2})$$

Consequently, by the Central Limit Theorem

$$\sqrt{n}(\widehat{\theta}_n^{(m)} - \widetilde{\theta}^{(m)}) \xrightarrow{\mathcal{D}} \mathcal{N}\left(0, \operatorname{Var}\left[-H_{\widetilde{\theta}^{(m)}}^{-1}\,\nabla_\theta L(S^{(m)}, \widetilde{\theta}^{(m)}) + D^{(m)}\operatorname{vec}(P^{(m)*})\right]\right),$$

where $D^{(m)} = (\partial\,\theta(F_0, P^{(m)}(F_0))/\partial\,\operatorname{vec}(P^{(m)}))^\top$. $\qquad\square$

*Proof of Corollary 5.3.* Similar as shown in Theorem 5.2, the original estimator satisfies the following expansion

$$\widehat{\theta}_n = \widetilde{\theta} + \frac{1}{n}\sum_{i=1}^{n} \theta^*(S_i) + o_p(n^{-1/2})$$

$$= \widetilde{\theta} + \frac{1}{n}\sum_{i=1}^{n} H_{\widetilde{\theta}}^{-1}\nabla_\theta L(S_i, \widetilde{\theta}^{(m)}) + o_p(n^{-1/2})$$

Then we have

$$(\widehat{\theta}_n^{(m)} - \widehat{\theta}_n) - (\widetilde{\theta}^{(m)} - \widetilde{\theta})$$

$$= \frac{1}{n}\sum_{i=1}^{n}\left[H_{\widetilde{\theta}}^{-1}\nabla_\theta L(S_i, \widetilde{\theta}^{(m)}) - H_{\widetilde{\theta}^{(m)}}^{-1}\nabla_\theta L(S_i^{(m)}, \widetilde{\theta}^{(m)}) + \left(\frac{\partial\,\theta(F_0, P^{(m)}(F_0))}{\partial\,\operatorname{vec}(P^{(m)})}\right)^\top \operatorname{vec}(P^{(m)*}(S_i))\right]$$

Therefore, by Central Limit Theorem, we have

$$\mathbb{E}(\|\widehat{\theta}_n^{(m)} - \widehat{\theta}_n\|^2) \leq \|\widetilde{\theta}^{(m)} - \widetilde{\theta}\|^2 + \frac{1}{n}\operatorname{Tr}\left(\operatorname{Var}\left[H_{\widetilde{\theta}}^{-1}G - H_{\widetilde{\theta}^{(m)}}^{-1}G^{(m)} + D^{(m)}\operatorname{vec}(P^{(m)*})\right]\right),$$

$\qquad\square$

*Proof of Theorem 5.4.* Consider the first-order Taylor expansion of $f(x;\theta)$

$$f(x;\widehat{\theta}_n^{(m)}) - f(x;\widetilde{\theta}^{(m)}) = \nabla_\theta f(x;\widetilde{\theta}^{(m)})^\top (\widehat{\theta}_n^{(m)} - \widetilde{\theta}^{(m)}) + o(\|\widehat{\theta}_n^{(m)} - \widetilde{\theta}^{(m)}\|).$$

From Theorem 5.2, we know $\|\widehat{\theta}_n^{(m)} - \widetilde{\theta}^{(m)}\| = O_p(n^{-1/2})$. Then, following the same expansion as shown in the proof of Theorem 5.2, along with Slutsky's theorem, gives the results. $\qquad\square$

## D  Limitations

The current work builds upon linear SDR, which assumes that both the target label and the sensitive attribute depend on linear projections of the representation. While this is a relatively mild assumption, it limits our ability to capture and mitigate nonlinear dependencies that may be embedded in the representation.

A promising future direction is to extend the framework using nonlinear SDR theory. In this approach, the representation is first mapped into a higher-dimensional feature space using a predefined kernel function, denoted by $\mathrm{Ker}(\boldsymbol{X})$. The goal is then to find a projection matrix $B_z$ such that

$$\boldsymbol{Z} \perp\!\!\!\perp \boldsymbol{X} \mid B_z^\top \mathrm{Ker}(\boldsymbol{X}),$$

enforcing conditional independence in the nonlinear space. Importantly, the core derivations and structural components of our method remain applicable in this generalized setting. Developing this extension is a promising direction for future research.

## E  Impact Statement

The framework proposed in this paper tackles the fundamental challenge of fairness in machine learning by directly mitigating bias in learned data representations. Departing from post-hoc de-biasing methods, we enforce fairness for new representations, enabling scalable and generalizable solutions for sensitive domains such as employment, healthcare, and finance. By balancing fairness and predictive utility through subspace decomposition, SFP helps reduce systemic disparities and promotes the development of trustworthy AI systems. This work underscores the importance of rigorous fairness evaluation and responsible deployment to advance equitable and accountable machine learning technologies.

