# OpenReview forum: "Understanding Fairness and Prediction Error through Subspace Decomposition and Influence Analysis"
_NeurIPS.cc/2025/Conference — NeurIPS 2025 poster_

### Official Review · Reviewer_MkjP · 2025-06-22

**Clarity:** 3
**Significance:** 3
**Originality:** 3
**Rating:** 5
**Confidence:** 3

**Summary:**

This paper proposes sequential fair projection, a fairness-aware learning framework that (i) decomposes feature space via sufficient dimension reduction into three orthogonal sub-spaces (Y-specific, Z-specific, Y-Z correlated) and (ii) controls fairness-utility trade-offs by incrementally adding shared components to the projection. Theoretical guarantees and numerical justifications are provided.

**Questions:**

(i) How to extend the current results to the more general $Z$ settings other than the current binary one?

(ii) The SDR procedure may be costly for a high-dimensional covariate $X$. What is the complexity of your algorithm?

(iii) Is it possible to derive a finite sample guarantee (asymptotic results are already provided in the paper)?

(iv) Please discuss potential intersectional fairness. For example, multiple sensitive attributes exist.

**Ethical Concerns:**

["NO or VERY MINOR ethics concerns only"]

**Final Justification:**

I suggest accepting the paper.

**Limitations:**

Yes

**Paper Formatting Concerns:**

No concern

**Quality:**

3

**Strengths And Weaknesses:**

Strengths: The paper is well-written and easy to follow. The theoretical results are rigorous, with comprehensive numerical results. And the idea is also novel.

Weaknesses:  (i) The sensitive variable is binary in the theoretical and numerical results; (ii) The SDR may be costly for high-dimensional X; (iii) Finite-sample performance is not discussed.

---

> ### Author Rebuttal · Authors · 2025-07-29
>
> We thank the reviewer for recognizing that our work is novel, well-motivated and grounded in solid theoretical foundations, and we appreciate the constructive feedback. We will address the remaining clarifications below point by point.
>
> ## Weakness
>
> Since the reviewer raised the same concerns in both the weaknesses and questions sections, we provide a consolidated response under the questions section only.
>
> ## Questions
>
> **Q1**: Our framework does not assume any specific structure for $Z$ in the model assumptions (Equation (3)). In particular, $Z$ is not restricted to being binary; the SDR framework naturally allows $Z$ to be scalar or vector-valued, and either categorical or continuous. However, since most existing fairness evaluation metrics (e.g., EO or DP) are defined for binary sensitive attributes, and most benchmark datasets contain only binary sensitive attributes, our experiments focus on that setting.
>
> **Q2**: The computational cost of obtaining the fair projection matrix primarily arises from estimating the candidate matrices $M_Y$, $M_Z$ and $M_{Y,Z}$, each typically constructed as a weighted covariance matrix. These computations scale linearly with the sample size $n$. In addition, the procedure includes an eigen-decomposition step for $p\times p$ matrices, and the rank estimation step scales linearly with dimension $p$.
>
> **Q3**: The finite sample analysis would be extremely challenging due to the chain of mappings described in Section 5.2, line 233. Existing theoretical results show that each individual step in the process achieves a $1/\sqrt{n}$ convergence rate to the population level, allowing us to drop higher-order terms in asymptotic analysis.
>
> However, for a finite sample analysis, it would require us to carefully characterize the error propagation across multiple stages of the estimation pipeline. This includes quantifying the cumulative effect of the estimation errors in the SDR subspaces, the projection matrices, and the final predictive model. Such an analysis would involve bounding the composition of errors and developing non-asymptotic guarantees that depend explicitly on the sample size, dimensionality, and possibly the eigenvalue gaps in the estimated subspaces. We recognize this as an important and nontrivial direction for future work.
>
> **Q4**: Our setting already accommodates intersectional fairness. When considering multiple sensitive attributes, we can simply concatenate them to form a multidimensional $Z$ without altering the model assumptions in Equation (3). This requires no changes to our framework or estimation procedure, and all asymptotic results remain valid. We will clarify and illustrate this point in the revised manuscript.
>
> **We appreciate your feedback, which has helped strengthen our work, and we hope our responses have addressed all of your concerns. Please let us know if there is any additional information or clarification we can provide to further support your understanding and confidence in our work.**

---

> > ### Comment · Area_Chair_HvN8 · 2025-08-05
> > **Reminder to follow up on author rebuttals**
> >
> > Hi Reviewer  MkjP,
> >
> > Please kindly follow up on the authors' rebuttal to your review, as we are approaching the end of the discussion period.
> >
> > Thanks.
> >
> > Regards,
> >
> > AC

---

> > ### Comment · Reviewer_MkjP · 2025-08-05
> > **Reply to Author Response**
> >
> > Thank you for your detailed response, which properly addressed all my concerns. I believe this is an interesting and high-quality paper that has made a significant contribution to the field.

---

> > > ### Author Response · Authors · 2025-08-05
> > >
> > > We sincerely thank the reviewer for the constructive feedback and continued support for our work, which
> > > have helped improve the quality and clarity of our final version.

---

### Official Review · Reviewer_6n9b · 2025-06-23

**Clarity:** 2
**Significance:** 3
**Originality:** 2
**Rating:** 4
**Confidence:** 3

**Summary:**

This work proposes a new approach to improve the fairness of algorithmic predictors by addressing the biases in the learned representations of the underlying data. The authors propose a novel approach using *sufficient dimension reduction* that decomposes the data space into sensitive attribute relevant, target variable relevant, and sensitive-target shared components. Using this decomposition, the work shows how the fairness-accuracy tradeoff can be navigated by leveraging these separate representation components. The authors provide a theoretical analysis and also empirically validate their approach on two popular fairness datasets.

**Questions:**

* How does the proposed approach's efficiency, specifically using SDR for dimension reduction, compare to other neural representation learning based approaches that look into disentanglement or fairness?
* Can the proposed approach with linear SDR decomposition also be applied in conjunction with a strong representation learning method for vision and textual settings?

**Ethical Concerns:**

["NO or VERY MINOR ethics concerns only"]

**Final Justification:**

The authors provide a new theoretically grounded application to improve fair representation learning that can maintain accuracy. After the rebuttal, I think the work is interesting, but the paper can be improved by having slightly more evaluations, and at a minimum, a more intuitive discussion of the theoretical results and analysis conducted.

**Limitations:**

Limitations are not discussed in the main paper. However, a brief discussion on technical limitations is provided in the appendix.

**Paper Formatting Concerns:**

None.

**Quality:**

3

**Strengths And Weaknesses:**

## Strengths
- The work provides a formal framework to study fairness from the perspective of representation subspaces by decomposing it into relevant subspaces. By having separate subspaces for sensitive features, target variable, and also having a subspace focused on the shared relations between the sensitive and target variables, the authors show how we can effectively navigate the accuracy fairness tradeoff.
- The approach is theoretically well-motivated, and the authors also provide a detailed theoretical analysis of their approach regarding predictive error and fairness tradeoffs. After using SDR to reduce dimensionality and finding the relevant subspaces, finding the most optimal fair predictor reduces to finding the feature subset that can provide high accuracy and low unfairness.
- Using synthetic data, the authors validate the theoretical results and show how increasing the number of shared components in the predictive process can improve the predictive performance *for all sensitive groups* but also *increase the performance gap* across the different groups, leading to potential unfairness.
- The authors also show how their method is effective in achieving good accuracy and fairness results in two real-world datasets (Adult and Bank).

## Weaknesses
- Since this work is highly related to prior works on *fair representation learning*, a more thorough discussion and comparison with more fair representation learning works would have been effective to position the authors' approach. Specifically, it would have been nice to have a discussion and also comparisons to approaches that have looked at fair representation learning, representation disentanglement, e.g., [1] and [2]. Works that have looked at *causal representation learning*, e.g., the work in [3], could also be relevant for a discussion and/or comparison.
- The author's approach relies on a linear SDR method, and it is unclear how well the method can work if it were necessary to extend it to more nonlinear or complex settings. To this end, the empirical evaluations are limited, focusing on only two tabular datasets without looking into image (e.g., CelebA) or text (e.g., BiasBios) datasets.
- It would be nice to have an understanding of the efficiency of the proposed method when compared to competing approaches mentioned by the authors, e.g., AdvDebias, and other deep-learning-based representation learning approaches cited here.

[1] Creager, Elliot, et al. "Flexibly fair representation learning by disentanglement." International conference on machine learning. PMLR, 2019.

[2] Louizos, Christos, et al. "The variational fair autoencoder." arXiv preprint arXiv:1511.00830 (2015).

[3] Yang, Mengyue, et al. "Causalvae: Disentangled representation learning via neural structural causal models." Proceedings of the IEEE/CVF conference on computer vision and pattern recognition. 2021.

---

> ### Author Rebuttal · Authors · 2025-07-29
>
> First, we thank the reviewer for recognizing that our work is well-motivated and grounded in solid theoretical foundations, and we appreciate the constructive feedback.
>
> Moreover, we would like to respectfully clarify that the primary concern regarding our use of linear SDR methods has already been acknowledged in the Limitations section. As discussed below, our work represents the **first principled attempt** to address the fairness–utility trade-off through a **theoretically grounded representation learning framework**, directly targeting key gaps in existing methods, namely, the lack of explicit control over fairness–accuracy trade-offs and the absence of theoretical guarantees. As is often the case with methodological contributions, it is expected that subsequent work will address the limitations of the initial framework. Indeed, we fully agree that extending the framework to nonlinear and large-scale domains (e.g., image or text data) would be a valuable direction for future work; we view these as natural and promising evolutions of our approach rather than shortcomings of the current contribution.
>
> We will address the remaining clarifications below point by point.
>
> ## Weakness
>
> **W1**: We thank the reviewer for pointing us to these additional references, which are highly relevant. However, we would like to clarify that our method differs significantly from these prior works, as outlined below:
>
> First, prior methods such as Creager et al. (2019) and Louizos et al. (2015) **lack explicit control** over the fairness–accuracy trade-off. These approaches impose fairness constraints through disentangled variational autoencoders and rely on tuning hyperparameters to achieve different levels of fairness and utility. However, they do not offer explicit guarantees or clear interpretability regarding how these trade-offs are managed. In contrast, our method is grounded in a statistical decomposition of the covariates $X$ into components that are predictive of the target label $Y$ and either orthogonal to or shared with the sensitive attribute $Z$. This results in a representation that is both **interpretable and transparent** on the information discarded and retained, **allowing for explicit and principled control** over the fairness–accuracy trade-off.
>
> Second, these prior methods rely on **optimization-based frameworks** involving variational autoencoders with adversarial or mutual information-based regularization. Such approaches are often sensitive to hyperparameter tuning and prone to convergence issues. In contrast, **our approach requires no model training, and is supported by theoretical guarantees**, thereby avoiding the instability, lack of transparency, and limited interpretability associated with adversarial learning.
>
> CausalVAE (Yang et al., 2021), while conceptually related in its goal of learning disentangled and causally interpretable representations, is not directly focused on fairness. Its framework is aimed at generative modeling through causal factorization and intervention, rather than fair prediction. When applied to fairness settings, CausalVAE would likely encounter the same two limitations noted above: **lack of explicit fairness–utility control and reliance on complex optimization procedures.**
>
> We will incorporate this discussion into the related work section of the revised paper.
>
> **W2**: We appreciate the reviewer’s insightful comment. It is true that our approach is built on a linear sufficient dimension reduction (SDR) framework, and as discussed in the Limitation, this assumption may limit applicability in highly nonlinear settings. However, our primary goal in this work is to develop a principled and interpretable framework for balancing fairness and utility through explicit statistical information decomposition.
>
> In line with our theoretical contribution, we evaluated the method on two widely used tabular datasets that allow for clear, interpretable assessments of fairness–utility trade-offs and are commonly used in fairness research. While our current implementation focuses on structured tabular data, we agree that evaluating the method on higher-dimensional and unstructured domains (such as CelebA or BiasBios) would be valuable. We view this as an important next step, and note that the subspace decomposition ideas we introduce could be extended using nonlinear SDR techniques (e.g., kernel or deep SDR), which we plan to explore in future work.
>
> Non-linear SDR techniques are indeed promising and are expected to outperform linear SDR methods, particularly on complex datasets such as those involving images and text. A straightforward extension would be to perform fair representation learning using neural network architectures that explicitly decorrelate the learned representations from sensitive variables. However, analyzing such models theoretically poses greater challenges. We believe this is a highly promising direction and plan to explore it in future work.
>
> **W3**: We are unclear about which sense of efficiency the reviewer is referring to, as statistical and computational efficiency are conceptually distinct.
>
> If the comment concerns ***statistical efficiency***, this typically refers to the convergence rate or asymptotic variance of an estimator. Our method achieves a convergence rate of $1/\sqrt{n}$ for all subspace estimators and post-SDR parameters. In contrast, most deep learning-based approaches lack theoretical guarantees on convergence or asymptotic behavior, making direct statistical efficiency comparisons infeasible.
>
> If the concern is about ***computational efficiency***, we note that our method avoids any optimization during subspace estimation. It relies only on matrix multiplications and closed-form calculations, enabling efficient execution on CPUs without the need for GPU acceleration. For instance, on the Adult and Bank datasets, our method takes 80.47 and 50.22 seconds, respectively, using only CPU. In comparison, deep learning-based methods such as AdvDebias, FairMixup, DRAlign, and DiffMCDP require iterative training and take approximately 110 and 90 seconds, while RLACE takes 155 and 127 seconds, respectively. This highlights that our method is more lightweight and accessible in terms of computational resources.
>
> ## Questions
>
> **Q1**: See our reply to **W3**.
>
> **Q2**: Yes, our method can be applied to high-dimensional settings, as SDR techniques are designed to extract low-dimensional structure from high-dimensional data. The performance of our approach depends on the structure of the original representation, and since our framework is limited to linear transformations, it may not fully capture non-linear dependencies. To address this, one could extend our method using non-linear SDR techniques such as kernel-based methods (as discussed in the appendix), or adopt information-theoretic approaches (e.g., mutual information) to better model non-linear relationships.
>
> We acknowledge that in complex settings, such as image and text embeddings, the assumptions of linear SDR may not hold, where our method may not be the most effective one. Nonetheless, the core idea of subspace disentanglement remains applicable. Developing a theoretical and interpretable non-linear extension is a promising yet challenging direction, which we will highlight in the revised manuscript.
>
> **We appreciate your feedback, which has helped strengthen our work, and we hope our responses have addressed all of your concerns. Please let us know if there is any additional information or clarification we can provide to further support your understanding and confidence in our work.**

---

> > ### Comment · Reviewer_6n9b · 2025-08-04
> >
> > I thank the reviewers for their detailed response. The computational efficiency aspect is interesting to know and improves the favorable applicability of the approach. I agree on the benefit of the approach focusing on pre-processing of the data to learn controllably fairer representations through decomposition. Pre-processing can be a very flexible approach and does away with the complexities of constrained optimization necessary for most in-processing approaches. However, I still feel that additional evaluations (without the theoretical grounding, that is understandable) could have helped. It is also unclear why the authors chose to control for DP in Algorithm 2, and whether that led to the incapacity to significantly reduce TPR differences. For the financial datasets studied, notions such as EOP and EOD are often more suitable. If Algorithm 2 is rerun with EOP or EOD, can that improve the results seen in the empirical evaluations?
> > It would also be very helpful if the authors could provide intuitive discussions and implications of the theoretical analysis provided.
> > Nonetheless, I think this is an interesting work and I am bumping my rating to 4.

---

> > > ### Author Response · Authors · 2025-08-05
> > >
> > > We thank the reviewer for the constructive feedback and continued support of our work.
> > >
> > > First, we would like to clarify that in Algorithm 2, we use Demographic Parity (DP) as an example of a fairness metric (see lines 209–210). However, the framework is flexible, users may replace DP with any other evaluation metric that suits their specific needs. In practice, as described in Section B.1 of the Appendix, we adopt MCDP (Maximum Cumulative Discrepancy of Predictive distributions) as our evaluation metric. Unlike DP, which compares group-wise expectations, MCDP captures the maximum discrepancy across the full predictive cumulative distribution functions between groups, making it a stricter and more informative measure. In fact, it is possible for a model with a lower DP to still have a higher MCDP, highlighting the stronger requirements MCDP imposes.
> > >
> > > We chose MCDP for evaluation because, in our experiments, the original dataset (using basic logistic regression) exhibits substantial unfairness under both DP and MCDP, while the TPR gap (associated with Equalized Odds) appears relatively small. This motivates our use of MCDP to better demonstrate the effectiveness of our method in achieving fairness.
> > >
> > > We will clarify these points in the final version to improve the clarity and completeness of the presentation.
> > >
> > > Once again, we sincerely thank the reviewer for the helpful suggestions to enhance the clarity and quality of our work.

---

### Official Review · Reviewer_Bsk4 · 2025-06-30

**Clarity:** 3
**Significance:** 2
**Originality:** 2
**Rating:** 4
**Confidence:** 4

**Summary:**

In the present work, the authors introduce Sequential Fair Projection (SFP), a representation-level debiasing framework based on linear sufficient dimension reduction (SDR): the approach entails (i) decomposing the feature space into label-only, sensitive-only, and shared subspaces; (ii) supplying a family of projection matrices that can re-inject up to m shared directions to trade off fairness and accuracy; (iii) analyzing the prediction error vs distance covariance-based unfairness performance. The practical recipe proposed by the authors is to pick the smallest shared sub-space projection whose validation accuracy is at least 95 % of the original model's performance. The authors validate SFP on synthetic data and two tabular benchmarks (Adult and Bank), where it attains competitive fairness with modest accuracy loss.

**Questions:**

On mixture of label/group information: Can the authors justify and empirically test the assumption that a linear projector might suffice when label and sensitive information are non-linearly entangled? What kind of issues could appear if there is a strong violation of this hypothesis?

On the chosen fairness-accuracy trade-off: Why choose the ad-hoc “≥ 95 % accuracy” cut-off? How sensitive are results to this hyperparameter? Is there some guarantee that one obtains the "fairest" model with that accuracy level?

On scalability in high-dimensional settings: What is the wall-clock cost of estimating SDR subspaces and looping over m on datasets at the scale of ImageNet, or transformer embeddings?

On the practical use of Theorems 5.2-5.4: How should a practitioner act on the given asymptotic variance bounds?

Evaluation scope – Why restrict experiments to two small tabular datasets with binary sensitive attributes? Have the authors tried multi-attribute settings or modern vision/NLP benchmarks?

Accuracy–fairness trade-off – On Adult, LR is far more accurate yet only moderately less fair. Could simpler post-hoc methods (e.g., threshold adjustment) close the gap?

Reproducibility – Code is promised, but hyper-parameters for SDR (slice count, rank estimation) and optimisation are missing from the main paper.

4. Limitations (as I see them)

**Ethical Concerns:**

["NO or VERY MINOR ethics concerns only"]

**Final Justification:**

With all the promised additions, I believe my main concerns have been addressed adequately in the rebuttal, and I am inclined to raise my score to a 4:borderline accept.

**Limitations:**

Some limitations are discussed in a paragraph at the end of the appendix. I believe that, in the spirit of transparency, they should be moved to the main text.

**Paper Formatting Concerns:**

See above.

**Quality:**

2

**Strengths And Weaknesses:**

**Strengths**
- The manuscript tackles fairness within representations, a worthwhile direction supported by clear notation and an SDR-based derivation.
- The presented influence-function analysis appears technically sound and may interest theory-minded readers.

**Weaknesses**
- The contribution feels somewhat incremental because prior work (e.g., Shi et al. 2024) already projects away sensitive subspaces, and iterative “add-back” heuristics are common in fairness fine-tuning.
- Methodologically, all results hinge on linear SDR projections, both for disentangling and for intersecting subspaces. This assumption is not necessarily realistic in modern representation learning, yet taken for granted in the main (a quick discussion is presented in the limitation section at the end of the appendix). Quantifying the degree of departure from this hypothesis in realistic settings would seem to be a necessary discussion topic in this work.
- The empirical evaluations appear limited: experiments cover only two small tabular datasets; no large-scale or pretrained-model study supports the broader narrative, or the competitiveness of the proposed method.
- The 95 % accuracy threshold and the feature-by-feature loop add a degree of arbitrariness and manual tuning (as many other methods that strike a balance between performance and fairness) and could largely increase computational cost in large-dimensional problems.
- There are some repetitions and typos, and some observables that would be useful for the reader to interpret the empirical evaluation of the proposed method are left in the appendix.

---

> ### Author Rebuttal · Authors · 2025-07-29
>
> First, we thank the reviewer for recognizing that our work is well-motivated and grounded in solid theoretical foundations, and we appreciate the constructive feedback.
>
> Moreover, we would like to respectfully clarify that the primary concern regarding our use of linear SDR methods has already been discussed in the Limitations section. As discussed below, our work represents the **first attempt** to address the fairness–utility trade-off through a **theoretically grounded representation learning framework**, closing the gaps in existing methods: lack of explicit control over fairness–accuracy trade-offs and the absence of theoretical guarantees. We fully agree that extending the framework to nonlinear setting would be a valuable direction for future work; we view these as natural and promising evolutions of our approach rather than shortcomings of the current contribution.
>
> We will address the remaining clarifications below point by point.
>
> ## Weakness
>
> **W1**: We appreciate the reviewer’s comments that our work shares a common tool, i.e., Sufficient Dimension Reduction, with Shi et al. (2024), and that related add-back methods also aim to achieve an intermediate level of fairness by balancing fairness and predictive accuracy. However, our contribution differs from these approaches in two important ways:
> 1. First, unlike Shi et al. (2024), which completely removes the influence of the sensitive attribute $Z$ from the representation of covariates $Z$ without leveraging label information $Y$, our method addresses a fundamentally different problem setup. We adopt a supervised approach that explicitly incorporates $Y$ to decompose the predictive information in $X$ into two components: (i) the part that is orthogonal to $Z$, and (ii) the part that is shared with $Z$. This label-informed decomposition enables more effective identification of useful predictive signal.
>
> 2. Second, while existing ``add-back’’ heuristics attempt to recover lost utility after fairness constraints, they are typically ad hoc and lack theoretical guarantees. In contrast, our approach formalizes the decomposition of shared and orthogonal information among $X$, $Y$ and $Z$, resulting in a method that is both interpretable and supported by statistically theory. And importantly, our principled formulation enables one to explicitly tune the fairness–accuracy trade-off while providing transparency on retained versus discarded information into the learned.
>
> **W2**: You are right that our proposed method is built on linear sufficient dimension reduction (SDR) projections, and as discussed in the Limitation section, violations of this assumption may affect its effectiveness. However, quantifying the degree of departure from this assumption is a non-trivial problem and depends on several factors, including the data distribution, signal-to-noise ratio, and the nature of the downstream task, as widely acknowledged in the SDR literature. We will clarify this point further and acknowledge it as an important direction for future research.
>
> **W3**: We appreciate the reviewer’s comment. Given the theoretical and methodological focus of our work, our goal in this paper is to present a principled framework rather than to conduct extensive, large-scale benchmarking. As is common in theory-oriented papers, we validate our approach on representative tabular datasets to illustrate the key properties and practical performance of the method. While we agree that evaluating on large-scale or pre-trained models would further support the broader narrative, we view this as a natural next step and plan to pursue such empirical validation in future work to complement the current theoretical contribution.
>
> **W4**: We thank the reviewer for this thoughtful observation. However, we would like to clarify that our method does not require the use of any accuracy threshold. The 95\% threshold is chosen to ensure a fair comparison across different methods. All methods and results reported in Table 1 follow this same criterion for hyperparameter tuning. Specifically, this threshold allows us to assess how fair the trained models are when maintaining a reasonable level of predictive performance. Users can easily adjust this threshold based on their specific tolerance for accuracy sacrifice in exchange for fairness.
>
> Regarding the feature-by-feature loop, it is designed to systematically identify and incorporate informative components by allowing a controlled level of unfairness, thereby achieving an intermediate point in the fairness-utility trade-off. This loop functions as a tuning parameter with improved interpretability. Unlike optimization-based methods that also require tuning a penalty coefficient $\lambda$, our approach offers explicit control and provides two main advantages: (1) the shared dimension is selected from a finite set of values, avoiding the need for grid search over a continuous $\lambda$ range, and (2) the interpretability of each added component allows practitioners to better understand the trade-off dynamics.
>
> Importantly, the number of shared dimensions is determined by the degree of entanglement between $Y$ and $Z$, rather than the dimensionality of the representation space. Thus, even in high-dimensional settings, the number of shared components may remain small.
>
> **W5**: We thank the reviewer for pointing out the repetitions, typos, and the placement of key observables in the appendix. We will carefully revise the paper to eliminate redundant text and correct all typos. We will move key summaries and metrics into the main text and clarify their relevance to the evaluation of our method.
>
> ## Questions
>
> **Q1**: Conditional independence tests can be employed to assess whether the estimated central subspaces are sufficient to capture all relevant information. For example, one can statistically test whether $Y \perp X \mid M_Y X$ and $Z \perp X \mid M_Z X$ hold. A number of existing works focus on this problem, such as the paper \textit{``Kernel-based Conditional Independence Test and Application in Causal Discovery'' (UAI 2011)}.
>
> If the linear SDR assumptions are strongly violated, then the estimated central subspaces may not adequately represent the conditional structure, and the matrices $M_Y$ and $M_Z$ may become full rank (i.e., rank $p$), indicating the need to include all dimensions. In such cases, we can still manually choose how many directions to retain in the post-SDR training based on the ordering of the eigenvalues, and form a fair projection accordingly.
>
> **Q2**: Our theoretical analysis and fairness utility trade-off does not rely on this 95\% threshold. It is intended solely to ensure a fair comparison between different methods. We will clarify this in our revised version.
>
> **Q3**: The computational cost of obtaining the fair projection matrix primarily arises from estimating the candidate matrices $M_Y$, $M_Z$ and $M_{Y,Z}$. These computations scale linearly with the sample size $n$. In addition, the procedure includes an eigen-decomposition step for $p\times p$ matrices, and the rank estimation step scales linearly with the ambient dimension $p$. For reference, the computation times for obtaining the all SDR subspaces $M_Y$, $M_Z$ and $M_{Y,Z}$ and their ranks on the Adult and Bank datasets are 80.47 seconds and 50.22 seconds only use CPU, respectively. We did not conduct additional experiments on large scaled image and contextual datasets as our work focuses on providing theoretical analysis and explanation, but it is an interesting task worth exploring for nonlinear SDR setting in the future work.
>
> **Q4**: Theorems 5.2--5.4 provide practitioners with statistical uncertainty quantification results for parameters estimated using the fair representation as well as their impact on predictive performance. These results enhance the interpretability of our framework by elucidating how the choice of dimension and the specific SDR technique employed influence the resulting parameter estimates. The practitioners can also benefit from constructing confidence intervals for the estimated parameters.
>
> **Q5**: The purpose of our simulation and real data experiments is to demonstrate that our proposed framework is effective in practice and that the underlying idea is sound. Our main contribution lies in the theoretical analysis and the insights it provides into the fairness-utility trade-off through the lens of subspace interactions.
>
> In more complex scenarios, such as those embeddings with intricate non-linear relationships, our linear SDR methods may be less effective in producing fair representations. Nonetheless, the core idea behind our approach remains applicable. By extending our framework to incorporate non-linear structures, e.g., through kernel-based SDR or information-theoretic formulation, we aim to close this gap in future work.
>
> **Q6**: Although the value of TPR gap is low for both the Adult and Bank datasets when using logistic regression, the DP and MCDP gaps remain high, indicating substantial group-level unfairness in terms of both expectations and distributions. Therefore, simply using post-hoc methods, such as threshold adjustment, can not address the underlying fairness issues.
>
> **Q7**: The number of slices is chosen as $p+1$, as stated in Section B.1. The parameters for rank estimation follow the same settings as presented in the original paper. For post-SDR training, we apply the baseline logistic regression model directly on the fair representations for both Adult and Bank datasets. We will clarify these implementation details in the revised version of the manuscript.
>
> **We appreciate your feedback, which has helped strengthen our work, and we hope our responses have addressed all of your concerns. Please let us know if there is any additional information or clarification we can provide to further support your understanding and confidence in our work.**

---

> > ### Comment · Reviewer_Bsk4 · 2025-08-05
> > **Comment**
> >
> > I would like to thank the authors for their thorough responses. I have some follow-up questions:
> >
> > > W1: ... In contrast, our approach formalizes the decomposition of shared and orthogonal information among $X$, $Y$ and $Z$, resulting in a method that is both interpretable and supported by statistically theory. And importantly, our principled formulation enables one to explicitly tune the fairness–accuracy trade-off while providing transparency on retained versus discarded information into the learned.
> >
> > I would challenge this, since the linearity assumption is not realistic, so some of these guarantees are lost in real-world applications. I understand that making further progress is necessary, in this direction, and not all problems need to be solved in the present work. But it is important not to overstate the practical applicability of this framework.
> >
> > > W2: ... We will clarify this point further and acknowledge it as an important direction for future research.
> >
> > I believe a (possibly small) empirical test in this direction would be a great addition to the present work. I understand that fully understanding the issues stemming from the violation of this hypothesis is beyond the scope of this manuscript. However, a preliminary discussion would solidify the presented results. For example, the approach presented in the answer to Q1, "we can still manually choose how many directions to retain in the post-SDR training based on the ordering of the eigenvalues, and form a fair projection accordingly", could be tested in some empirical setting.
> >
> > > W4: ... Unlike optimization-based methods that also require tuning a penalty coefficient $\lambda$, our approach offers explicit control and provides two main advantages: (1) the shared dimension is selected from a finite set of values, avoiding the need for grid search over a continuous $\lambda$ range, and (2) the interpretability of each added component allows practitioners to better understand the trade-off dynamics.
> >
> > I appreciate the importance of point (2). On point (1), instead, I would argue that the discrete nature of the selection would hardly lower the computational cost for achieving the selected bias-accuracy trade-off compared to a standard regularization approach.
> >
> > > Q3: ...  These computations scale linearly with the sample size $n$. In addition, the procedure includes an eigen-decomposition step for $p\times p$ matrices, and the rank estimation step scales linearly with the ambient dimension $p$.
> >
> > I believe this discussion should be included in the main of the paper, since the cubic scaling with $p$ could be problematic in large-scale datasets.

---

> > > ### Author Response · Authors · 2025-08-05
> > >
> > > We sincerely thank the reviewer for the constructive suggestions to improve our work.
> > >
> > > First, regarding the linear assumption, we fully agree that a dedicated discussion or remark in the main paper is necessary. We will clarify what this assumption entails, under what conditions it may be violated, and how one might proceed when such violations occur. To support this, we will include additional empirical results in the synthetic data section. Specifically, we will demonstrate that when the linearity assumption is violated, one must manually select the appropriate number of post-SDR dimensions based on the spectrum of eigenvalues.
> > >
> > > Second, concerning the shared dimension selection, we would like to clarify that we did not claim this approach would reduce computational cost compared to standard regularization methods. However, it should not be significantly worse. Even in high-dimensional settings, a grid search over shared dimensions is still feasible. For example, we can iterate over every 2 or 3 dimensions due to the ordered nature of eigenvalues, or even adopt a bisection search strategy. We will include this clarification in a remark in the final version.
> > >
> > > Lastly, on the topic of computational complexity, we will add a remark to discuss it. While eigenvalue decomposition has a theoretical complexity of $O(p^3)$, there exist efficient numerical algorithms and optimized libraries that significantly accelerate this process in practice. Moreover, since all candidate matrices in our case are real and positive definite, the decomposition is more efficient than general cases. To illustrate this, we performed a simple experiment using Python's internal function ***np.linalg.svd*** under the same computational environment used in our experiments. The average running time over 10 repetitions for various dimensions $p$ is summarized in a table and shows the decomposition is computationally feasible even at larger scales.
> > >
> > > | **Dimension \(p\)**       | 100     | 200     | 400     | 800     | 1600    | 3200    | 6400     |
> > > |---------------------------|---------|---------|---------|---------|---------|---------|----------|
> > > | **Avg SVD Time (s)**      | 0.00297 | 0.01379 | 0.04137 | 0.14096 | 0.58752 | 2.85565 | 35.65711 |
> > >
> > > **Table**: Average time to perform SVD on $p \times p$ real positive definite matrices (10 repetitions)
> > >
> > > We hope our response addresses your concerns, and we would be happy to provide further explanation or clarification on any remaining questions.

---

> > > > ### Comment · Reviewer_Bsk4 · 2025-08-06
> > > > **Reply**
> > > >
> > > > I thank the authors for their response. With all the promised additions, I believe my main concerns have been addressed adequately in the rebuttal, and I am inclined to raise my score to a 4:borderline accept.

---

### Official Review · Reviewer_L3zY · 2025-07-03

**Clarity:** 3
**Significance:** 3
**Originality:** 3
**Rating:** 5
**Confidence:** 4

**Summary:**

In this work the authors argue to look at fairness through representational decomposition, analyzing the separate subspaces
of a model representation that contain information pertaining to a sensitive attribute, to a target variable, and to both. Their
analysis provides a method of interpolating between maximum utility (and worst fairness) and minimum utility (and best
fairness) by removing more and more of the shared subspace between the two variables. Through extensive theoretical
analysis, they demonstrate the utility of their decomposition as a way to debias representations and propose an algorithm to
create fair predictors.

**Questions:**

* How do these estimators scale with data and representation space size? In your setting you focused on dimensionality
of 16 in the synthetic setting and of a similar scale for Adult. However, some of your feature embedding baselines scale
up to LLM feature spaces on the order of hundreds if not thousands of dimensions. Would this method be competitive there?

**Ethical Concerns:**

["NO or VERY MINOR ethics concerns only"]

**Final Justification:**

I think this paper will be a good contribution to NeurIPS and deserves an accept. The authors answered my questions during the review process as well and communicated clearly about the message and motivation of the work.

**Limitations:**

The authors have not discussed the limitations, but I do not think there are any critical points missing.

**Paper Formatting Concerns:**

* In line 147, is $B$ equivalent to $M_Y$?
 * Maybe use a different rv than U for the full sufficient representation to not confuse it with a uniform.
 * Line 226,“omit the using the” is a typo

**Quality:**

4

**Strengths And Weaknesses:**

## Strengths
 * The paper provides rigorous theoretical analysis of their approach, demonstrating that their approximation results in non-
increasing error for the Bayes predictor as they include more of the shared subspace. Additionally, the authors have
experiments (both synthetic and on real-world data) which validate the efficacy of their approach on established fairness
metrics.

## Weaknesses
 * In the predictive settings evaluated in the paper, there are many existing (strong) methods for fairness. In fact, part of your
argument is that moving beyond predictive intervention is important. I think a key piece of evidence missing in this paper is a
setting where predictive interventions fail, or where representational fairness is important.
 * Additionally , you may want to look to work on privacy and specifically the privacy funnel [1], where a similar task is at hand:
how can we modify X such that it provides no information about a private attribute Z while maintaining maximum utility for Y.

[1] Calmon, F. P., Makhdoumi, A., & Médard, M. (2015, June). Fundamental limits of perfect privacy. ISIT 2015

---

> ### Author Rebuttal · Authors · 2025-07-29
>
> We thank the reviewer for recognizing that our work is well-motivated and grounded in solid theoretical foundations, and we appreciate the constructive feedback. We will address the remaining clarifications below point by point.
>
> ## Weakness
>
> **W1**: We do not claim that predictive approaches are inadequate or must be entirely replaced. Instead, our contribution lies in offering additional insights into achieving intermediate levels of fairness and providing interpretability for the fair representations we construct. While the predictive setting is the most commonly used approach for achieving fairness, typically by incorporating a fairness penalty term during training, it often suffers from optimization instability. This is because the prediction loss and the fairness loss can be inherently conflicting, leading to convergence issues and fluctuating training dynamics. Moreover, predictive fairness methods are usually task-specific: if one wishes to apply the learned model to a different downstream task or to enforce a different type of fairness constraint, it often requires retraining the model from scratch.
>
> In contrast, our approach focuses on representational fairness, which seeks to remove information about the sensitive attribute directly from the learned representation. A key advantage of this approach is that it offers a clear and interpretable trade-off between fairness and utility: by controlling the number of dimensions, we can balance fairness constraints against predictive performance. Moreover, once such a fair representation is obtained, it can be reused across multiple downstream tasks or fairness definitions without requiring retraining. This flexibility makes representational fairness particularly appealing in multi-task or evolving fairness scenarios, as it decouples fairness enforcement from task-specific prediction objectives.
>
> We will incorporate this discussion into the related work section in the revised version of the manuscript.
>
> **W2**: We thank the reviewer for suggesting this reference. Indeed, it shares a similar motivation, which aims to ensure privacy by minimizing the information content related to sensitive variables in the learned representation. This aligns well with our fairness perspective, and it opens an interesting direction for extending our framework. Specifically, we can consider non-linear transformations of representations and formalize fairness objectives using dependence measures such as mutual information or distance covariance. For example, by minimizing the dependence between the representation and the sensitive attribute (e.g., minimizing $I(X,Z)$ while preserving utility (e.g., maximizing $I(X,Y)$, we can construct fair representations in a more general, potentially non-linear setting. We will incorporate this perspective into the conclusion and future work section of our revised manuscript.
>
> ## Questions
>
> **Q1**: The computational cost of obtaining the fair projection matrix primarily arises from estimating the candidate matrices $M_Y$, $M_Z$ and $M_{Y,Z}$, each typically constructed as a weighted covariance matrix. These computations scale linearly with the sample size $n$. In addition, the procedure includes an eigen-decomposition step for $p\times p$ matrices, and the rank estimation step scales linearly with dimension $p$. Notably, our method remains applicable in high-dimensional settings, as SDR techniques are specifically designed to extract low-dimensional structure from high-dimensional data.
>
> As for performance, it largely depends on the structure of the original representation. Since our current framework focuses on linear transformations, it may fail to capture non-linear dependencies. One potential solution is to extend our approach using non-linear SDR techniques, such as kernel-based methods, as discussed in the limitations section of the appendix. Alternatively, as mentioned in our response to your comment on weaknesses, we can explore information-theoretic approaches (e.g., mutual information) to handle non-linear relationships in the representation learning process.
>
> ## Paper Formatting Concerns
>
> We thank the reviewer for pointing out the typos and potential confusion notations, we will revise them in the final manuscript. In line 147, $B$ is the standardized eigenvectors of $M_Y$, which makes $BB^{\top}$ a projection matrix without losing any information relates to $Y$.
>
> **We appreciate your feedback, which has helped strengthen our work, and we hope our responses have addressed all of your concerns. Please let us know if there is any additional information or clarification we can provide to further support your understanding and confidence in our work.**

---

> > ### Comment · Reviewer_L3zY · 2025-08-04
> > **Reviewer Response**
> >
> > Thank you for answering my questions! I understand many of the downsides of predictive fairness interventions and agree that your method offers new insights and angles to tackle the fairness problem from. My main point is that your work would have much more impact if you had an example demonstrating how fairness at a representational level unlocks new applications that predictive fairness cannot tackle. Your theory is very nice but your experiments are limited to synthetic settings and Adult, where we already have many fairness interventions we know work well to debias predictors, so the potential impact of the work is not as well demonstrated as it could be. Overall I think this paper will be a good contribution to NeurIPS and I'll be keeping my score.

---

> > > ### Author Response · Authors · 2025-08-05
> > >
> > > We sincerely thank the reviewer for the constructive feedback and continued support for our work.
> > >
> > > In the final version, we will include a motivating example to illustrate how the limitations of predictive fairness can be addressed by considering fairness at the representation level. Briefly, while predictive fairness enforces fairness in the model's output, it does not prevent the representation or raw data from being highly correlated with the sensitive attribute. Enforcing fairness at the representation level tackles the issue at its root and may also provide broader benefits for dataset developers and government agencies by guiding the release of fairer data.
> > >
> > > Regarding the experiments, we fully understand your concern. Since this work represents our first step toward developing a new analytical framework, we focus our efforts and space on clearly presenting the theoretical foundations. In future work, we plan to extend the framework to nonlinear settings and explore more complex datasets and experiments to further demonstrate the trade-off between fairness and utility in practical applications.
> > >
> > > Once again, we thank you for your valuable feedback and thoughtful discussion, which have helped improve the quality and clarity of our final version.

---

### Decision · Program_Chairs · 2025-09-17

**Decision:**

Accept (poster)

**Comment:**

This paper proposes a fairness framework that uses sufficient dimension reduction to decompose feature representations into target-relevant, sensitive, and shared components, thereby allowing explicit control of the fairness–utility trade-off. Reviewers found the work theoretically rigorous, noting the influence-function analysis and theoretical guarantees as strengths. They appreciated the clarity of the problem formulation and the interpretability of the trade-off.

Concerns were raised about the limited scope of the experiments, which were restricted to synthetic data and two tabular datasets, and about the reliance on linear SDR, which may not hold in more complex or nonlinear settings. Reviewers also pointed out that the contribution feels somewhat incremental given related work, and that the fairness–accuracy trade-off involved ad hoc choices such as the 95% accuracy threshold. Some reviewers requested additional discussion of related work on fair representation learning and privacy.

In the rebuttal, the authors clarified that their goal is to complement predictive fairness methods by offering representational fairness with interpretable trade-offs. They emphasized computational efficiency, provided further detail on how fairness metrics can be adapted, and acknowledged the linear SDR assumption as a limitation while suggesting extensions to nonlinear settings. Reviewers responded positively to these clarifications, with some maintaining their accept scores and others raising to borderline accept, while still noting that broader evaluations would strengthen the work.

Overall, reviewers agreed that the theoretical contribution is solid and the paper is well-motivated, despite the limited empirical scope. After the rebuttal, all reviewers supported acceptance, with ratings in the accept to borderline accept range. I therefore recommend acceptance.